# Impact of frontal ablation on the ice thickness estimation of marine-terminating glaciers in Alaska

Beatriz Recinos[1,2], Fabien Maussion[3], Timo Rothenpieler[1], and Ben Marzeion[1,2]

[1]Institute of Geography, Climate Lab, University of Bremen, Bremen, Germany
[2]MARUM - Center for Marine Environmental Sciences, University of Bremen, Bremen, Germany
[3]Department of Atmospheric and Cryospheric Sciences, Universität Innsbruck, Innsbruck, Austria

**Correspondence:** B. Recinos (recinos@uni-bremen.de)

**Abstract.**

Frontal ablation is a major component of the mass budget of calving glaciers, strongly affecting their dynamics. Most global scale ice volume estimates to date still suffer from considerable uncertainties related to i) the implemented frontal ablation parameterisation or ii) not accounting for frontal ablation at all in the glacier model. To improve estimates of the ice thickness distribution of glaciers, it is thus important to identify and test low-cost and robust parameterisations of this process. By implementing such parameterisation into the ice-thickness estimation module of the Open Global Glacier Model (OGGM v1.1.2), we conduct a first assessment of the impact of accounting for frontal ablation on the estimate of ice stored in glaciers in Alaska. We find that inversion methods based on mass conservation systematically underestimate the mass turnover, and therefore the thickness of tidewater glaciers when neglecting frontal ablation. This underestimation can amount to up to 19 % on a regional scale and up to 30 % for individual glaciers. The effect is independent of the size of the glacier. Additionally, we perform different sensitivity experiments to study the influence of i) a constant of proportionality ($k$) used in the frontal ablation parameterisation, ii) Glen's temperature-dependent creep parameter ($A$) and iii) a sliding velocity parameter ($f_s$) on the regional dynamics of Alaska tidewater glaciers. OGGM is able to reproduce previous regional frontal ablation estimates applying a number of combinations of values for $k$, Glen's A and $f_s$. Our sensitivity studies also show that differences in thickness between accounting for and not accounting for frontal ablation occur mainly at the lower parts of the glacier, both above and below sea level. This indicates that not accounting for frontal ablation will have an impact on the estimate of the glaciers' potential contribution to sea-level rise. Introducing frontal ablation increases the volume estimate of Alaska marine-terminating glaciers from $9.18 \pm 0.62$ to $10.61 \pm 0.75$ mm SLE, of which $1.52 \pm 0.31$ mm SLE ($0.59 \pm 0.08$ mm SLE when ignoring frontal ablation) are found to be below sea level.

# 1 Introduction

Estimates of the spatial distribution of ice thickness are needed as initial conditions for glacier models, for attempting to understand how glaciers respond to climate change, and for quantifying their contribution to sea-level rise. Despite this importance, ice thickness measurements around the globe are scarce, performed only in approx. 600 glaciers (Gärtner-Roer et al., 2014) out
of more than 200,000 identified in the latest Randolph Glacier Inventory (RGI v6.0, Pfeffer et al., 2014). In order to overcome this under-sampling problem, a number of methods have been developed to infer the total volume and/or the ice thickness distribution of glaciers from characteristics of the glacier surface properties. Some of these methods rely on scaling approaches relating the length, slope and area of the glacier to its total volume (e.g. Bahr et al., 1997; Lüthi, 2009; Radić and Hock, 2011; Grinsted, 2013). Others rely on parameterisations of basal shear stress (e.g. Paul and Linsbauer, 2012; Linsbauer et al., 2012;
Frey et al., 2014), on observed surface velocities (e.g. Gantayat et al., 2014), or on applying the shallow-ice approximation (e.g. Oerlemans, 1997; Cuffey and Paterson, 2010) and/or an integrated form of Glen's flow law (see Farinotti et al. (2017), for a review of all these methods and Farinotti et al. (2019), for a global-scale intercomparison).

One method presented by Farinotti et al. (2009) and successfully applied several times since then (e.g. Morlighem et al., 2011; Huss and Farinotti, 2012; Clarke et al., 2013; Maussion et al., 2019), combines ice flow dynamics and mass conservation
principles to constrain mass fluxes through given glacier cross-sections. The method infers ice thickness from estimates of ice fluxes derived from the assumption that ice fluxes balance the surface mass budget (Farinotti et al., 2009). The results are thus sensitive to the spatial distribution of the mass flux and the mass balance. For calving glaciers, the surface mass budget cannot be considered balanced, even assuming equilibrium between glacier and climate. The derived ice thickness estimate for these glaciers hence depends on estimates of frontal ablation.

Frontal ablation (mass loss by calving and frontal melting Pope, 2012), is an efficient process to deliver ice from glaciers and ice sheets into the ocean. It has contributed substantially to sea-level rise in the past and played an important role in the stability of ice sheets and tidewater glaciers during the Pleistocene (Benn et al., 2007). Calving is strongly coupled with dynamical processes inside the glacier. An increase in the ice flux can trigger a calving event and in turn this event can accelerate the movement of the ice. External aspects like ocean temperature, fjord bathymetry and, in polar areas, sea-ice concentration
along the calving front can also influence the discharge of solid ice to the ocean (Straneo et al., 2013). As a consequence of the diverse nature of calving processes, the development of parameterisations of frontal ablation in numerical ice sheet and glacier models remains an important challenge. There is a wide spectrum of approaches that vary in scale and complexity, justified through the diversity of intended applications of the models (Price et al., 2015).

There have been many successful efforts to represent frontal ablation for individual glaciers (e.g. Ultee and Bassis, 2016;
Åström et al., 2014; Todd and Christoffersen, 2014; Oerlemans et al., 2011; Nick et al., 2010). While these achieve encouraging results, it is unlikely that they can be implemented in a global glacier model anytime soon, because of the amount and quality of data needed to constrain this type of model. The crevasse-depth criterion proposed by Nick et al. (2010) for example, requires knowledge of surface melt and refreeze rates at the crevasses of the glacier tongue, and crevasse depth observations to calibrate and validate these rates. These kinds of observations are hard to obtain for entire glaciated regions: e.g., the 198 calving glaciers

in Alaska investigated here, or the 3,222 glaciers classified as calving (marine- and lake- terminating) glaciers in the RGI v6.0. Other recent calving models that use discrete particles or a full-Stokes model approach (e.g. Åström et al., 2014; Todd and Christoffersen, 2014; Todd et al., 2018) are too computationally expensive to be included in global glacier models that seek to consistently simulate past and future global scale glacier changes.

At the regional and global scale, very few estimates of frontal ablation fluxes of glaciers outside the ice sheets exist (Blaszczyk et al., 2009; Burgess et al., 2013; McNabb et al., 2015; Huss and Hock, 2015). From all the global glacier models published in the last decade, only Huss and Hock (2015) account for frontal ablation of marine-terminating glaciers. However, this model, along with the rest of ice thickness inversion methods, still suffers from considerable uncertainty associated with the uncertainty of the frontal ablation parameterisation.

For improving ice thickness distribution estimates at the global scale, it is thus important to identify and test low-cost and robust parameterisations of frontal ablation that might not resolve all the dynamical processes at the calving front (e.g. subaqueous frontal melting, subaerial frontal melting and sublimation), but that can estimate the amount of ice passing through the terminus of the glacier during a mass balance year. Using the ice-thickness estimation module of the Open Global Glacier Model (OGGM v1.1.2), we assess the impact of frontal ablation on the estimate of ice stored in Alaska glaciers classified as

marine-terminating in the RGI v6.0 (also referred to tidewater glaciers in this study).

Alaska glaciers cover approximately 12 % of the global glacier area outside of the ice sheets (Kienholz et al., 2015). In the RGI (v6) there are 27109 glaciers in the region occupying an area of 86776.6 $km^2$, including adjacent glaciers in the Yukon and in British Columbia. From these glaciers, 51 have been classified as marine-terminating (74 km of tidewater margin) and 147 as lake- and river-terminating glaciers (420 km of lake/river margin) occupying an area of 11962.4 $km^2$ and 16720.6 $km^2$,

respectively. Calving glaciers (marine- and lake- terminating) occupy approximately 33 % of the Alaska glacier area (Fig. 1; Pfeffer et al., 2014; Kienholz et al., 2015).

The glaciers are divided into six subregions in the RGI. Subregions 1 and 3 contain only land terminating glaciers. Calving glaciers are mostly concentrated in the subregions 4, 5, and 6, along the mountain ranges of the southern Alaska coast (Fig. 1), an area characterised by maritime climate and topography reaching > 5000 $ma.s.l$ (Kienholz et al., 2015). Glaciers contained

in the RGI in this region range in size from a few square kilometres (Ogive Glacier, 2.8 $km^2$) to many thousands of square kilometres (Hubbard Glacier, 3400 $km^2$; McNabb and Hock, 2014).

The subregions 4 and 5 are well studied glacierised areas of Alaska. McNabb et al. (2015) presented a 28 year record (1985 - 2013) of frontal ablation for a subset of marine-terminating glaciers that includes the 27 most dominant tidewater glaciers of the region. They represent 96 % of the total tidewater glacier area in the gulf of Alaska. The total mean rate of frontal

ablation was estimated to be 15.11 $\pm$ 3.63 Gt $yr^{-1}$ (16.48 $\pm$ 3.96 $km^3$ $yr^{-1}$), over the period 1985 - 2013. Other studies also reported similar values (e.g. Larsen et al., 2007). Frontal ablation in this region is heavily dominated by two glaciers in particular: Hubbard and Columbia Glaciers (McNabb et al., 2015). Additionally, McNabb et al. (2015) identified 36 actively calving tidewater glaciers in Alaska; 27 of those were used to estimate the total mean rate of frontal ablation presented in McNabb et al. (2015).

We implement a simple parametrisation of frontal ablation in OGGM, following the approaches proposed by Oerlemans and Nick (2005) and Huss and Hock (2015). By performing sensitivity studies on the model, we i) investigate the effect of accounting for frontal ablation on the ice thickness estimation of OGGM and on the ice volume estimate for these glaciers, and ii) study the impact of varying several OGGM parameters (the calving constant of proportionality $k$, Glen's temperature-dependent creep parameter $A$, and sliding velocity parameter $f_s$) on the regional frontal ablation rates of Alaska.

## 2  Input data and pre-processing

### 2.1  Glacier outlines and local topography

The glacier outlines used in this study are those defined in the region 1 of the RGIv6. Four glaciers (Columbia, Grand Pacific, Hubbard and Sawyer Glacier) were merged with their respective pair branches (West Columbia, Ferris, Valerie and West Sawyer Glacier) into a single outline. A local map projection is defined for each glacier in the inventory following the methods described in Maussion et al. (2019). A Transverse Mercator projection is used, centred on the glacier in order to conserve distances, area and angles. Then, topographical data is chosen automatically depending on the glacier's location and interpolated to the local grid. For this study we used a combination of the Shuttle Radar Topography Mission (SRTM) 90 m Digital Elevation Database v4.1 (Jarvis et al., 2008) for all latitudes below $60^\circ$N and the Viewfinder Panoramas DEM3 product (90 m, http://viewfinderpanoramas.org/dem3.html) for higher latitudes. For Columbia Glacier, we used the DEM from the Ice Thickness Models Intercomparison eXperiment (Farinotti et al., 2017, ITMIX) instead [1]. All datasets are re-sampled to a resolution depending on glacier size (Maussion et al., 2019) and smoothed with a Gaussian filter of 250 m radius.

### 2.2  Glacier flowlines, catchment areas and widths

The glacier centrelines are computed following an automated method based on the approach of Kienholz et al. (2014). Fig. 2a illustrates an example of this geometrical algorithm applied to the Columbia Glacier. The centrelines are then filtered and slightly adapted to represent glacier flowlines with a fixed grid spacing (Fig. 2c). The geometrical widths along the flowlines are obtained by intersecting the normals at each grid point with the glacier outlines and the tributaries' catchment areas. Each tributary and the main flowline has a catchment area, which is then used to correct the geometrical widths. This process assures that the flowline representation of the glacier is in close accordance with the actual altitude-area distribution of the glacier. The width of the calving front, therefore, is obtained from a geometric first guess multiplied by a correction factor. This may lead to uncertainties in the frontal ablation computations, as discussed in Sect. 5.

---

[1] See Sect. 5 for a discussion about the importance of reliable topographic data for the frontal ablation estimate.

## 2.3 Regional frontal ablation estimates

Frontal ablation for 27 marine-terminating glaciers presented by McNabb et al. (2015) are used to compare the results of the model and calibrate the calving constant of proportionality $k$. These estimates were calculated from satellite-derived ice velocities and modeled estimates of glacier ice thickness.

## 2.4 Climate data and mass balance

The mass balance (MB) model implemented in OGGM uses monthly time series of temperature and precipitation. The current default is to use the gridded time-series dataset CRU TS v4.01 (Harris et al., 2014), which covers the period of 1901-2015 with a $0.5^{\circ}$ resolution. This raw, coarse dataset is downscaled to a higher resolution grid (CRU CL v2.0 at 10' resolution, New et al., 2002), following the anomaly mapping approach described in Maussion et al. (2019), allowing OGGM to have an elevation-dependent climate dataset from which the temperature and precipitation at each elevation of the glacier are computed, and then converted to the local temperature according to a temperature gradient (default: $6.5$ K km$^{-1}$). No vertical gradient is applied to precipitation, but a correction factor $p_f =2.5$ is applied to the original CRU time series (see Maussion et al., 2019, appendix A for more information). The MB model (see Sect. 3.2) is calibrated with direct observations of the annual surface mass balance (SMB). For this, OGGM uses reference mass-balance data from the World Glacier Monitoring Service (WGMS, 2017) and the links to the respective RGI polygons assembled by Maussion (2017).

## 3 Open Global Glacier Model (OGGM) and frontal ablation parameterisation

For this study, a simple frontal ablation parametrisation is implemented into the Open Global Glacier Model (OGGM v1.1.2). OGGM is developed to provide a global scale, modular and open source numerical model framework for consistently simulating past and future global scale glacier change. The mathematical framework of the model and its capabilities have been explained in detail by Maussion et al. (2019). In this section, we will only describe the modifications done to the mass-balance and ice thickness inversion modules, together with the frontal ablation parametrisation implemented in order to improve the initialisation of the model for marine-terminating glaciers. Sect. 3.3 provides details on the limitation of applying the parameterisation to lake-terminating glaciers.

### 3.1 Ice thickness

The method of estimating ice thickness from mass turnover and principles of ice-flow dynamics in glaciers go back to Budd and Allison (1975) and Rasmussen (1988), whose ideas were further developed by Fastook et al. (1995) and Farinotti et al. (2009). The later aims to estimate ice thickness distribution from a given glacier surface topography, which can be achieved assuming that the mass-balance distribution should be balanced by the ice-flux divergence. This method has been modified in OGGM in order to implement a new ice thickness inversion procedure physically consistent with the flowline representation of the glaciers and taking advantage of the mass-balance calibration procedure of OGGM (see below).

The flux of ice $q$ (m$^3$ s$^{-1}$) through a glacier cross-section of area $S$ (m$^2$) is defined as:

$$q = \bar{u}S \tag{1}$$

with $\bar{u}$ being the average cross-section velocity (m s$^{-1}$). By applying the well known shallow-ice approximation (Hutter, 1981, 1983; Cuffey and Paterson, 2010; Oerlemans, 1997) and making use of the Glen's ice flow law, we compute the depth-integrated centerline velocity $u$ of the cross-section with:

$$u = \frac{2A}{n+2} h_0 \tau^n \tag{2}$$

with $A$ being the ice creep parameter (which has a default value of $2.4 \times 10^{-24}$ s$^{-1}$ Pa$^{-3}$), $n$ the exponent of Glen's flow law (default: $n=3$), $h_0$ the centerline ice thickness (m), and $\tau$ the basal shear stress defined as:

$$\tau = \rho g h_0 \alpha \tag{3}$$

with $\rho$ the ice density (900 kg m$^{-3}$), $g$ the gravitational acceleration (9.81 m s$^{-2}$) and $\alpha$ the surface slope (computed along the centerline). Optionally, a sliding velocity $u_s$ can be added to the deformation velocity to account for basal sliding, using the following parametrisation (Oerlemans, 1997; Budd et al., 1979):

$$u_s = \frac{f_s \tau^n}{h_0} \tag{4}$$

with $f_s$ a sliding parameter (default: $5.7 \times 10^{-20}$ m$^{-2}$s$^{-1}$ Pa$^{-3}$). We then assume that the centerline velocity is equal to the average section velocity ($\bar{u} \approx u$), which in absence of lateral drag is correct for a rectangular bed shape but isn't in the parabolic case, where we neglect the variations of the shear stress (and $u$) along the parabola. In the parabolic case and with N=3, this results in a section velocity overestimation of a factor 315 / 128 (approx 2.46) in comparison to the section velocity obtained by integrating the shallow-ice velocity over the parabola. We proceed with this approximation because (i) this factor cannot be computed analytically for any other non-integer value of Glen N or for other bed shapes (e.g. trapezoidal) and (ii) the uncertainties about the true shape of the bed would make the model very sensitive to this choice. The computed flux in OGGM however does vary by a factor 2/3 depending on whether one assumes a parabolic ($S = \frac{2}{3} hw$) or rectangular ($S = hw$, with w being the glacier width) bed shape. The default in OGGM is to use a parabolic bed shape, unless the section touches a neighbouring catchment or neighbouring glacier (ice divides, computed from the RGI). For the last five grid points of tidewater glaciers, the bed shape is also assumed to be rectangular. Singularities with flat areas are avoided since the constructed flowlines are not allowed to have a local slope $\alpha$ below a certain threshold (default: 1.5°, see Maussion et al., 2019).

Following the approach described in Maussion et al. (2019), $q$ can be estimated from the mass-balance field of a glacier. If $u$ and $q$ are known, $S$ and the local ice thickness $h$ (m) can also be computed by making some assumptions about the geometry of the bed and by solving Eq. 1. This equation becomes a polynomial in $h$ of degree 5 with only one root in $\mathbb{R}_+$, easily computable for each grid point.

## 3.2 Mass-balance and ice flux $q$

OGGM's mass balance model is an extension of the model proposed by Marzeion et al. (2012) and adapted in Maussion et al. (2019), to calculate the mass balance of each flowline grid point for every month, using the CRU climatological series as boundary condition. The equation governing the mass-balance is that of a traditional temperature index melt model. The monthly mass-balance $m_i$ (kg m$^{-2}$ s$^{-1}$) at elevation $z$ is computed as:

$$m_i(z) = p_f P_i^{\text{solid}}(z) - \mu^* \, max \, (T_i(z) - T_{\text{melt}}, 0) \tag{5}$$

where $P_i^{\text{solid}}$ is the monthly solid precipitation, $p_f$ a global precipitation correction factor, $T_i$ the monthly temperature and $T_{\text{melt}}$ is the monthly mean air temperature above which ice melt is assumed to occur (default: -1°C). Solid precipitation is computed as a fraction of the total precipitation: 100 % solid if $T_i <= T_{solid}$ (default: 0°C), 0 % if $T_i >= T_{\text{liquid}}$ (default: 2°C), and linearly interpolated in between. The parameter $\mu^*$ indicates the temperature sensitivity of the glacier, and it needs to be calibrated: in a nutshell, the MB calibration consists of searching a 31-year climate period in the past during which the glacier would have been in equilibrium while keeping its modern-time geometry, implying that the mass balance of the glacier during that period in time $m_{31}(t)$ is equal to zero, with $m_{31}(t)$ being the glacier integrated mass-balance computed for a 31 yr period centred around the year $t$ (e.g. $t^* = 1962$ for most glaciers in Alaska) and for a constant glacier geometry fixed at the RGI outline's date (e.g. 2009 for the Columbia Glacier). It should be noted that the mass balance calibration in OGGM excludes MB measurements from tidewater glaciers as reference data, for reasons described below.

This "equilibrium mass-balance" ($m_{31}(t)$) is then assumed to be equal to the "apparent mass-balance" ($\tilde{m} = \dot{m} - \rho \frac{\partial h}{\partial t}$) as defined by Farinotti et al. (2009), where the flux of ice $q$ through a glacier catchment area ($\Omega$) is defined as:

$$q = \int_\Omega (\dot{m} - \rho \frac{\partial h}{\partial t}) dA = \int_\Omega m_{31} dA \tag{6}$$

If the glacier is land-terminating, $\int m_{31} = 0$ by construction (a property which is used to calibrate $\mu^*$ in Eq. 5). $q$ is then obtained by integrating the equilibrium mass-balance $m_{31}$ along the flowline(s). $q$ starts at zero and increases along the major flowline, reaches it's maximum at the equilibrium line altitude (ELA) and decreases towards zero at the tongue (Maussion et al., 2019).

However, this assumption does not hold for tidewater glaciers, where a steady state implies that:

$$\int m_{31} = \frac{q_{calving} \, \rho}{A_{RGI}} \tag{7}$$

Where $q_{calving}$ is the frontal ablation flux of the glacier (m$^3$ yr$^{-1}$). This flux is then converted to units of specific MB (kg m$^{-2}$ yr$^{-1}$) by multiplying with the ice density (900 kg m$^{-3}$) and dividing by the total glacier area as given by the RGI. A more precise definition would be that $q_{calving}$ is the average amount of ice that passes through the glacier terminus in a year for a glacier in equilibrium with the climate forcing. This has direct consequences for the calibration of the temperature sensitivity

parameter $\mu^*$. With all other things kept equal, two otherwise identical glaciers (one calving, one non-calving) will have to have different temperature sensitivities $\mu^*$: the calving glacier will have a lower $\mu^*$, resulting in a lowered Equilibrium Line Altitude (ELA), a positive surface mass budget, and finally to a mass flux through the terminus. The objective here is to allow the model to have a non-zero calving flux, with the goal of improving the glacier thickness inversion computed by OGGM.

5 ### 3.3 Frontal ablation parameterisation

### 3.3.1 Calving law

To account for frontal ablation of marine-terminating glaciers we employ a calving law proposed by Oerlemans and Nick (2005) and that has already been applied at a large scale by Huss and Hock (2015). The annual frontal ablation flux $q_{calving}$ ($km^3 \ yr^{-1}$) is computed as a function of the height $(h_f)$, width $(w)$ and estimated water depth $(d)$ of the calving front as:

$$q_{calving} = max(0; kdh_f) \cdot w \tag{8}$$

$k$ is a calibration parameter (which has a default value of $2.4 \ yr^{-1}$ in this study). The water depth $(d)$ is estimated from free-board, using elevation, and ice thickness $(h_f)$ data obtained from the model output:

$$d = h_f - E_t + z_w \tag{9}$$

Where $E_t$ is the elevation of the glacier surface at the terminus and $z_w$ is the elevation of the water body with respect to sea
level. The water depth $(d)$ is estimated using the terminus elevation $(E_t)$ obtained by projecting the RGI outline onto the DEM (i.e., the terminus elevation is the top of the cliff). We follow the same definition as Oerlemans and Nick (2005) where $d$ is the bed elevation with respect to sea level. For lake-terminating glaciers, we are not able to estimate a water depth since one would need to know the free-board of the glacier terminus, i.e. the elevation of the glacier lake surface. For this reason, most of our experiments and results focus on marine-terminating glaciers only ($z_w$ is set of 0 m a.s.l.), with the exception of the experiment
presented in section 4.2.

Unlike Huss and Hock (2015), who estimated the thickness of the calving front $(h_f)$ by scaling approaches, we solve for the ice thickness by prescribing that the amount of ice calved ($q_{calving}$) must be equal to the amount of ice delivered to the terminus by OGGM ($q$, computed from ice deformation and sliding in Section 3.1):

$$q_{calving} = q \tag{10}$$

$q_{calving}$ varies with $h_f$ as a polynomial of degree 2. $q$ is a polynomial in $h_f$ of degree 5 (with n = 3 in Eq. 2), with an extra term in degree 3 if we account for a sliding velocity (see Eq. 4). Eq. 10 is therefore a polynomial that can be solved for $h_f$.

### 3.3.2 Illustration of the method

We use the LeConte Glacier (see Fig. 2b and d) as a test case to illustrate our solution method. Fig 2d shows the result of the model's default ice thickness inversion procedure, which assumes an ice flux of zero at the terminus ($q_{calving} = 0$). Note that
by default, the ice thickness at the glacier front $h_f$ is zero.

First we examine how the frontal ablation flux ($q_{calving}$) from the calving law would change if we increase the terminus ice thickness of the glacier, while keeping the free-board fixed ($E_t$ is the only variable known in Eq. 9 "with certainty", from the DEM surface elevation at the terminus). Fig. 3a shows that the flux remains equal to zero as long as $h_f$ is not thick enough to reach water, after which the water depth is positive and calving occurs. At this point, we are unaware of the real frontal ablation flux for this glacier, but we make some very coarse assumptions:

- Oerlemans and Nick (2005) calving law is perfectly exact

- the tuning parameter $k$ is known

- our glacier is in equilibrium with climate (we assume mass-conservation inversion in OGGM)

- ice deformation at the glacier terminus follows Glen's flow law

Under these assumptions, we set up an experiment where we compute a frontal ablation flux (from the calving law, Eq.8) for a range of prescribed frontal ice thicknesses (see Fig. 3b, blue line), then give this flux back to the inversion model which computes a frontal ice thickness according to the physics of ice flow (Fig. 3b, green and orange lines). As shown in Fig. 3b, both curves meet at a frontal thickness value which complies with both the calving law ($q_{calving}$) and the ice thickness inversion model of OGGM ($q$). Note that changing Glen's deformation parameter $A$ or adding sliding does not change the problem qualitatively: we will still solve a polynomial degree 5 in OGGM, with a new term in degree 3.

Fig. 3c displays the same data as Fig. 3b (here as a function of the prescribed water depth), showing more clearly that there are two locations where the zero line is crossed and the condition of Eq. 10 is met. However, only one solution (the larger one) provides a realistic water depth, and therefore a realistic frontal ablation flux.

### 3.3.3 Implementation

We solve the polynomial in Eq. 10 numerically, via bound-constrained minimisation methods (algorithm provided by SciPy Jones et al., 2001), which leads to a quick convergence. The advantage over an analytical solution is that numerical solvers have the flexibility to be applied to any other formulation of $q_{calving}$ and $q$, i.e. that this method will still be applicable if a lateral drag parameterisation or another formulation for the calving law is added to OGGM in the future.

After finding the solution for the frontal ice thickness ($h_f$) and the corresponding frontal ablation flux ($q_{calving}$), we give this flux back to the mass balance model (Eq. 7), adjust the temperature sensitivity of the glacier $\mu^*$, and invert for a new ice thickness distribution for the entire glacier (see Sect. 4.1 for results). This always results in a adjustment of $\mu^*$ towards lower values in order to lower the ELA and unbalancing the steady-state surface mass-budget for a frontal ablation flux to exist. Note that this adjustment of mass-balance is always necessary (regardless of the choice of model parameters such as $k$ or Glen's $A$) in order to ensure mass-conservation and the update of upstream ice thickness.

However, sometimes the flux estimated by the calving law (Eq. 8) is too large to be sustained by the surface mass-balance. Even without glacier melt ($\mu^* = 0$), the total accumulation over the glacier is too small to close the frontal mass budget. This can be due to several factors: frontal ablation is overestimated, or solid precipitation is underestimated. The frontal ablation can be

overestimated if $k$ and/or the calving law does not represents the dynamics of that particular glacier, or if $h_f$ is overestimated. In most cases (see results), it is possible to find a realistic $\mu^*$ compatible with a frontal ablation flux, but when this is not possible $\mu^*$ is fixed to zero and the frontal ablation flux $q_{calving}$ is obtained by closing the mass-budget instead of using the calving law.

## 4 Results

We apply this frontal ablation parameterisation to all marine-terminating glaciers in Alaska. We study the impact of including this parameterisation on the estimated glacier thickness, volume and ice flow velocity. The following sections describe different sensitivity experiments: i) varying the frontal ablation flux added to the MB model and assessing the impact on glacier volume, ii) varying several model parameters (Glen's flow law ice creep parameter $A$, a sliding parameter $f_s$, and the calving constant of proportionality $k$) and assessing each parameter's impact to the regional frontal ablation of Alaska, and iii) show the impact of different model configurations (obtained from the sensitivity experiments of sect. 4.4) to the total volume of Alaska marine-terminating glaciers. The parameter set up for each configuration can be found in Table 1.

### 4.1 Case study: Columbia Glacier

The Columbia Glacier located in south-central Alaska, is one of the most studied tidewater glacier in the world. With a detailed record of its retreat since 1976, it is the single largest contributor of the Alaska glaciers to sea-level rise (Berthier et al., 2010; Larsen et al., 2015). The ice flow, ice discharge and tidewater retreat of the glacier are all extensively documented, providing rich insight into the underlying processes that modulate tidewater glacier behaviour and stability (McNabb et al., 2012). These reasons motivated the selection of the Columbia Glacier as an exemplary study site to illustrate our results for an individual glacier, while the goal of our approach is the ability to improve the model representation of any calving glacier.

Following the process described in section 3.3, we calculate a virtual frontal ablation for the Columbia Glacier of 2.98 km$^3$ yr$^{-1}$ (2.73 Gt yr$^{-1}$). This flux represents the estimated amount of ice passing through the terminus of the glacier, if the glacier was in equilibrium with the climate for a constant glacier geometry fixed at the RGI outline's date (e.g. 2009 for this glacier). This estimate was obtained using the model's default values for the parameters $k$, $A$ and $f_s$. McNabb et al. (2015) estimated a mean frontal ablation of $3.53 \pm 0.85$ Gt yr$^{-1}$ during 1982–2007, with previous studies estimating 5.5 Gt yr$^{-1}$ for the same period (Rasmussen et al., 2011). Fig. 4 shows the difference between not accounting for frontal ablation in the mass balance ($q_{calving} = 0$ in Eq. 7) and accounting for frontal ablation, adding the frontal ablation flux calculated to the MB module ($q_{calving} = 2.98$ km$^3$ yr$^{-1}$ in Eq. 7). If $q_{calving} = 0$ (Fig. 4a), we estimate the total volume of the Columbia Glacier to be 270.40 km$^3$, 29.21% less than the volume calculated if the frontal ablation is added (Fig. 4b), which results in a volume of 349.39 km$^3$.

When computing the ice thickness distribution map of the glacier, the impact of accounting for frontal ablation is mainly reflected in the two adjacent branches of the Columbia Glacier (Fig. 4b) and at the glacier terminus (Fig. 4c). An overview of the glacier main centreline profile is shown in Fig. 4c, together with the 2007 thickness map published by McNabb et al. (2012)

(green dotted line), a study that provided a reconstructed bed topography and ice thickness, based on velocity observations of the Columbia Glacier and mass conservation. Fig. 4c also includes the result of the "consensus estimate" for the Columbia glacier ice thickness from Farinotti et al. (2019). OGGM's glacier bed estimation without accounting for frontal ablation (grey line) as well as the composite solution from Farinotti et al. (2019) (yellow line) estimate zero thickness at the calving front.

By accounting for frontal ablation in OGGM's MB and thickness inversion modules, we can compute a bedrock profile closer to the 2007 bed map, especially close to those points located at the terminus of the glacier. The frontal ablation parameterisation allows OGGM to grow a thick calving front at the glacier terminus. Additionally, we observe that both bed estimations from OGGM (grey and black lines, Fig. 4c) diverge primarily below sea level.

## 4.2    Frontal ablation and glacier volume

In this experiment, we assign a frontal ablation flux ranging from $0 - 5 \, \mathrm{km}^3 \, \mathrm{yr}^{-1}$ to each glacier classified as potentially calving in the RGI v6.0, keeping the model's default values for the parameters $A$ and $f_s$. The aim is to calculate the changes in volume for each glacier as a function of the frontal ablation value, while keeping the rest of the aspects that control the volume of the glacier fixed to the default values (e.g. solid precipitation, outline, topography, ice parameters). As a result of the automated workflow of OGGM, we are able to calculate the changes in volume of all 198 calving glaciers in Alaska[2], for each value in
the frontal ablation flux range.

The results of this experiment are shown in Fig. 5, where the frontal ablation fluxes are expressed as a fraction of the annual accumulation ($p_f P_i^{Solid}(z)$) over each individual glacier. This fraction is de facto normalised to a maximum of 1, since the calving flux cannot exceed the total accumulation. Large glaciers (green and red lines in Fig. 5) won't reach this value in the prescribed calving range of $0 - 5 \, \mathrm{km}^3 \, \mathrm{yr}^{-1}$. Eq. 5 and 7 indicate that a temperature sensitivity $\mu^* \leq 0$ would imply that the
glacier is producing a frontal ablation larger than its annual accumulation. When this happens, OGGM clips the temperature sensitivity $\mu^*$ to zero, setting a physical limit to the frontal ablation of each individual calving glacier.

Fig. 5a shows that the effect of frontal ablation on the glacier volume is systematic, in that accounting for frontal ablation in the MB will always result in an increase of the glacier volume. Even if the frontal ablation fraction is only 0.14 of the total accumulation, a glacier volume can be underestimated by up to 20% if we ignore this extra source of ablation. However,
there is a wide range of sensitivities of the estimated glacier volume to the calving flux, and no simple relation to e.g. glacier size was found. Other glacier specific parameters likely to play a role are the slope, the accumulation area ratio and the total precipitation.

## 4.3    Effect of frontal ablation on ice velocity

To analyse the effect of frontal ablation on ice velocity, we keep the same model configuration (default values of $k$, $A$ and
$f_s$) and calculate the average ice velocity along the main flowline for all marine-terminating glaciers that produced a frontal ablation flux. Fig. 6 shows the difference between the average velocity output of the model when accounting for frontal ablation

---

[2]Only in this section we include lake-terminating glaciers in the experiments, because we are not calculating a frontal ablation flux but assigning a specific value to the mass-balance equation (Eq. 7).

and without accounting for frontal ablation. When taking frontal ablation into account, the glaciers experience an increase in ice velocity towards the terminus. This increase of velocities is due to an increase of the mass flux (and therefore ice thickness) when we account for frontal ablation.

These results highlight the importance of applying a frontal ablation parameterisation at the initialisation stages of the model in order to recreate a realistic tidewater glacier behaviour. Without this extra term on the mass balance, velocities and ice thicknesses go to zero towards the terminus. This is not only a problem for the inversion procedure: these unrealistic features will also affect the dynamical runs realised with the forward model, i.e. any calving parameterisation applicable in the future will rely on a realistic bedrock to work properly.

Note that these velocities are not surface velocities but average section velocities. Annual surface velocities would have to be estimated from these values and the vertical profile of velocity in order to be compared to observations. Furthermore, since we run OGGM under an equilibrium assumption, the results presented here will not reflect the transient states that appear in observations. The usefulness of any comparison to other data (observed or modeled) is therefore limited. Additionally, some of the velocity maps in Alaska previously published (e.g. Burgess et al., 2013) are computed with many glaciers undergoing significant interannual velocity variability over the observation interval and only one velocity snapshot is included in the maps. Velocities might thus not represent long term average. However, comparing surface velocities derived from OGGM with observations might be useful when no previous estimates of frontal ablation fluxes exist in a RGI region (e.g. Greenland), providing another way of calibrating OGGM parameters.

## 4.4 Sensitivity studies in Alaska marine-terminating glaciers

We perform different sensitivity experiments to study the influence of i) the calving constant of proportionality $k$, ii) Glen's temperature-dependent creep parameter A and iii) sliding velocity parameter, on the regional frontal ablation of Alaska. The results of these experiments are shown in Fig. 7. In the first experiment we vary the calving constant of proportionality $k$ in a range of 0.24 - 2.52 $\mathrm{yr}^{-1}$ and used the model default values for Glen A and sliding parameter. Fig. 7a shows that our estimate for the regional frontal ablation matches the regional estimate by McNabb et al. (2015) if $k$ has an approximated value of 0.63 $\mathrm{yr}^{-1}$, in the case of excluding sliding ($f_s = 0$), or if $k$ is equal to 0.67 $\mathrm{yr}^{-1}$ in the case of including a sliding velocity (with $f_s = 5.7{\times}10^{-20} \mathrm{\ s}^{-1} \mathrm{\ Pa}^{-3}$). It is important to emphasize that the regional frontal ablation from McNabb et al. (2015) only comprises 27 glaciers but that they represent an estimated 96% of the total frontal ablation of Alaska.

We then keep these two values of $k$ and vary the values of Glen $A$ creep parameter and sliding parameter ($f_s$). The results are shown in Fig. 7 b and c. It is well known that ice flow models are sensitive to the values chosen for parameters describing ice rheology and basal friction (e.g. Enderlin et al., 2013; Brondex et al., 2017). As expected, our frontal ablation estimates are also sensitive to different values of Glen $A$ and sliding parameter, but highly dependent on different values of $k$, at least for the first part of the $k$ values range (0.24 - 0.80 $\mathrm{yr}^{-1}$). The linear relationship between $q_{calving}$ and $k$ at the start of the curve in Fig. 7a is mainly a consequence of the calving law used in the parameterisation. For larger $k$ values ($\geq 0.8 \mathrm{\ yr}^{-1}$) the shape of the curve is due to OGGM's physical constraint of clipping $\mu^*$ to zero and calculating the maximum $q_{calving}$ allowed by the local climate (see Sect.3.3.3).

Maussion et al. (2019) showed that both sliding and ice rheology ($A$) have a strong influence on OGGM's computed ice volume, hence a strong influence on the thickness of the glacier and in this case the frontal ablation estimate. Like in Maussion et al. (2019); Fig. 7 shows that one could always find an optimum combination of Glen A and sliding parameters that lead to (in this case) previously calculated frontal ablation estimates. Enderlin et al. (2013) also showed that when such flowline models are applied to a tidewater glacier, there is a non-unique combination of these parameter values that can produce similar stable glacier configurations, making $k$, Glen $A$ and $f_s$ parameters highly dependent on observations of either frontal ablation, ice velocity or glacier ice thickness.

## 4.5 Regional volume of marine-terminating glaciers for different model configurations

Finally, we compute the total volume of marine-terminating glaciers for different "equally good" parameter sets based on the results of Fig. 7 a, b and c. Each configuration is constructed by finding the intercepts between the model frontal ablation estimates and the regional estimate from McNabb et al. (2015), including the intercepts to the lower and upper error (see Fig. 7). A summary of the different parameter sets used for each model run can be found in Table 1 and the results of each configuration are shown in Fig. 8. Each configuration was run twice: once setting $q_{calving} = 0$, then a second time accounting for frontal ablation.

Similarly to the results shown in sections 4.1 and 4.2, Fig. 8 shows that there are significant differences between total volume estimates without and with accounting for frontal ablation. Volume estimates after accounting for frontal ablation are 11.7 to 19.7% higher than the volume estimates ignoring frontal ablation, considering all model configurations shown in Table 1, indicating a robust relationship. We find that there are no significant differences between the resulting volumes for different $k$ values and that the differences in volume estimates between configurations are mainly due to adding or ignoring a sliding velocity or varying the value of the Glen $A$ creep parameter.

Additionally, we also calculate the regional ice volume below sea level. The results for Columbia Glacier discussed in section 4.1 might create the impression that the differences in thickness along the main centerline, with and without accounting for frontal ablation, are not relevant for the potential glacier contribution to sea-level rise, since most of the differences in thickness (grey and black line in Fig. 4c) are found below sea level. However, Fig. 8 shows that considering the whole region, a significant fraction of the total volume difference is found above sea level, implying that accounting for frontal ablation will directly impact the estimate of these glacier's potential contribution to sea-level rise. By introducing frontal ablation, the volume estimate of marine-terminating glaciers in Alaska is increased from $9.18 \pm 0.62$ to an average of $10.61 \pm 0.75$ mm SLE, of which $1.52 \pm 0.31$ mm SLE are found to be below sea level (instead of only $0.59 \pm 0.08$ without). The uncertainties presented here are the standard deviation of the model configurations shown in Tab. 1. The consensus estimate from Farinotti et al. (2019) for these glaciers is 7.68 mm SLE, 27.58% lower than our average estimate of $10.61 \pm 0.75$ mm SLE.

## 5 Discussion

We have shown that the model is capable of computing regional frontal ablation estimates by tuning model parameters with published regional-scale estimates of frontal ablation, but the question of model performance for individual glaciers still remains open. In areas with no observational data or previous knowledge of frontal ablation, OGGM could make use of physical constraints (e.g. that $\mu^*$ must be greater than zero) as well as bathymetry and terminus width estimates to calibrate the model at the glacier scale. In the following section, we will explain such calibrations, discuss other parameters that affect frontal ablation estimates and discuss these estimates for individual glaciers.

In all previous model runs, we used the standard OGGM terminus geometry computation without correcting the width and water depth at the glacier front using potentially known values from other sources. As a result, not all of the glaciers classified as marine-terminating glaciers in the RGI produce a frontal ablation flux in OGGM (6 glaciers). This is mainly due to a wrong estimation of the water depth from free-board. These glaciers typically have a high terminus elevation (e.g. $E_t = 151.96$ m. a.s.l for the Chenega Glacier, RGI60-01.09639), for which the only possible value of $q_{calving}$ that complies with both the calving law and the ice thickness inversion model of OGGM is a $q_{calving} = 0$, since there is not enough mass turnover to grow a calving front under our mass conservation assumptions (see Sect. 3.3.2). The wrong water depth estimation can thus be best explained by a poor surface altitude estimation at the calving front ($E_t$). The problematic surface altitude estimation in turn can probably be explained by a mismatch between the acquisition dates of the DEM and the glacier outline.

Maussion et al. (2019) noted that a number of glaciers will suffer from poor topographic information, especially those located in the high latitudes. Most marine-terminating glaciers are located in regions where cloud free satellite measurements are rare. Therefore, the DEM of these regions might present errors (e.g. A wrong elevation at the terminus and/or a date of data acquisition which does not match that of the RGI outlines) that will spread to water depth estimations from free-board (see Eq. 9). The possibility of using higher resolution DEM's such as the ArcticDEM was explored during this study but was quickly eliminated because of large data voids present on the data, especially for big glaciers (e.g. Hubbard Glacier). However, new data sets such as the TanDEM-X (Wessel et al., 2018) are currently being explored for future versions of OGGM.

For 36 marine-terminating glaciers, we assess the model performance in comparison to the estimates by McNabb et al. (2015), with and without corrections for these errors (Fig. 9a). Calving front widths were corrected with the Alaska Tidewater Glacier Terminus Positions database (McNabb and Hock, 2014). The database contains terminus positions for 49 marine-terminating glaciers. Since three of these glaciers (Grand Pacific, Hubbard and Sawyer Glacier) were merged with their respective pair branches (Ferris, Valerie and Sawyer western Glacier), we are left with a total of 46 glacier terminus widths. The widths are computed by selecting the terminus positions closest to the glacier's RGI outline and by averaging the widths that resulted from the projection of the vector lines selected. These widths are used to correct OGGM's flowline width at the calving front in the cases where the model is not able to represent the real calving front width. The last flowline width at the front of the glacier is then clipped to the width value estimated from the database. To smooth the transition between the clipped value and OGGM's flowline width, we linearly interpolate between the clipped value and 5 pixels upstream on the flowline.

We then correct the modified widths to preserve the same glacier area than the RGI's. By doing this, we slightly modify the altitude-area distribution of the glacier.

Additionally, multi-beam bathymetry data from NOAA National Centers for Environmental Information (2004) was used to estimate the water depth in front of the glacier terminus. This data was used only for glaciers where the DEM resolution would not allow an estimate of the water depth from elevation data and ice thickness (free-board). The bathymetry data was compiled into a raster format and provided to us by Robert McNabb (pers. comm.). Both corrections were used for Fig. 9 only.

Fig. 9a demonstrates that without calibrating any OGGM parameter (only using the model default values for Glen A, $f_s$ and $k$), but making use of additional data (e.g. terminus positions and bathymetry), we are able to estimate a frontal ablation flux for individual glaciers within the same order of magnitude as those estimated by McNabb et al. (2015). The model root mean square error (RMSE) is reduced from 1.08 km$^3$ yr$^{-1}$ (mean deviation of 0.28) to 0.53 km$^3$ yr$^{-1}$ (mean deviation of 0.11). Even though part of these errors may arise from the fact that glaciers are in a disequilibrium state at the time of the McNabb et al. (2015) estimate, errors in boundary conditions (e.g., topography date not coinciding with the glacier outline date and uncertainties in the frontal width) and plain model errors also contribute. By using bathymetry and real terminus width estimates we improve the boundary conditions of the parameterisation that are highly dependent on the DEM quality.

When these corrections (terminus width and water depth) are not implemented and errors occur while estimating the real terminus geometry, OGGM has to rely on clipping $\mu^*$ to be larger than or equal to zero, setting a physical limit where the frontal ablation flux for each individual tidewater glacier cannot be larger than its annual accumulation ($p_f P_i^{Solid}(z)$). This is not ideal, because it implies that all of the glacier's ablation in an equilibrium setting is due to frontal ablation and no surface melt occurs, which is unrealistic in the climate conditions of Alaska. For applications on the global scale, bathymetry data and terminus mapping will be very valuable in regions with poor topographic resolution and where no observations of frontal ablation exist.

Additionally, we compare our final glacier volume estimates with the default configuration and by correcting the terminus geometry. These results are shown in figure Fig. 9b, together with the consensus volume estimate for each glacier from Farinotti et al. (2019). Fig. 9b shows that even if these corrections to the glacier terminus might have a large effect on the frontal ablation flux for some glaciers (e.g. Hubbard glacier), the effect is not as big as if we do not account for frontal ablation at all. This is the case of the consensus estimate from Farinotti et al. (2019) (red bars in Fig. 9b), where the models used do not account (or crudely account) for this extra lost of mass when inverting for the ice thickness.

## 6 Conclusions

We have implemented a frontal ablation parameterisation into OGGM and shown that inversion methods ignoring frontal abla-tion systematically underestimate the mass flux and thereby the thickness of calving glaciers. Accounting for frontal ablation in ice thickness inversion methods based on mass conservation (as listed in Farinotti et al., 2017) increases estimates of the regional ice mass stored in marine-terminating glaciers by approximately 11 to 19 %. While for individual glaciers, ice volume may be underestimated by up to 30% when ignoring the impact of frontal ablation, the effect is independent of the size of the

glacier. Implementing a frontal ablation parameterisation allows OGGM to represent a non-zero thickness calving front, which is necessarily the case when no ice flux is assumed to cross the glacier terminus. This parameterisation is key for initialising the glacier's thickness in the model.

The model was able to reproduce previously calculated regional frontal ablation estimates by finding the best combination of values for $k$, Glen's $A$ and the sliding parameters. Note that this comparison is limited by the equilibrium condition imposed on OGGM during initialisation, which is not the case in observations. The best-performing parameter set for transient runs of OGGM may be different.

Our sensitivity studies also show that the differences in thickness, between adding or not frontal ablation to the MB model, occur mainly at the lower parts of the glacier, but often above sea level. This indicates that not accounting for frontal ablation will have an impact on the estimate of this glacier's potential contribution to sea-level rise.

Additionally, our experiments highlight the need for bathymetry data and terminus mapping, as they may constrain model parameters when the DEM quality is not sufficient to provide a realistic estimate of the terminus geometry.

*Code availability.*

The OGGM software together with the frontal ablation parameterisation module are coded in the Python language and licensed under the GPLV3 free software license. The latest version of the OGGM code is available on Github (https://github.com/OGGM/oggm), the documentation is hosted on ReadTheDocs (http://oggm.readthedocs.io), and the project webpage for communication and dissemination can be found at http://oggm.org. The code and data used to generate all figures and analyses of this paper can be found at https://github.com/bearecinos/cryo_calving_2019. The OGGM version used for this study is available in a permanent DOI repository (https://doi.org/10.5281/zenodo.2580277).

*Author contributions.*

BM and FM are the initiators of the OGGM project and conceived this study. BR is the main developer of the frontal ablation parameterisation module and wrote most of the paper. FM is the main OGGM developer and was largely involved with the development of the frontal ablation parameterisation module. TR made significant contributions to the OGGM code and the frontal ablation parameterisation module and is responsible for the successful deployment of the code on supercomputing environments.

*Acknowledgements.* BR was supported by the DFG through the International Research Training Group IRTG 1904 ArcTrain. We would like to thank Robert McNabb for providing the Columbia Glacier thickness map and the bathymetry data which he compiled into raster formats. We also thank Chris Miele for discovering issues with the code and for helping extracting the water depth from the bathymetry raster files. We thank the editor Etienne Berthier, Douglas Brinkerhoff and an anonymous referee for their thoughtful reviews of this paper.

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

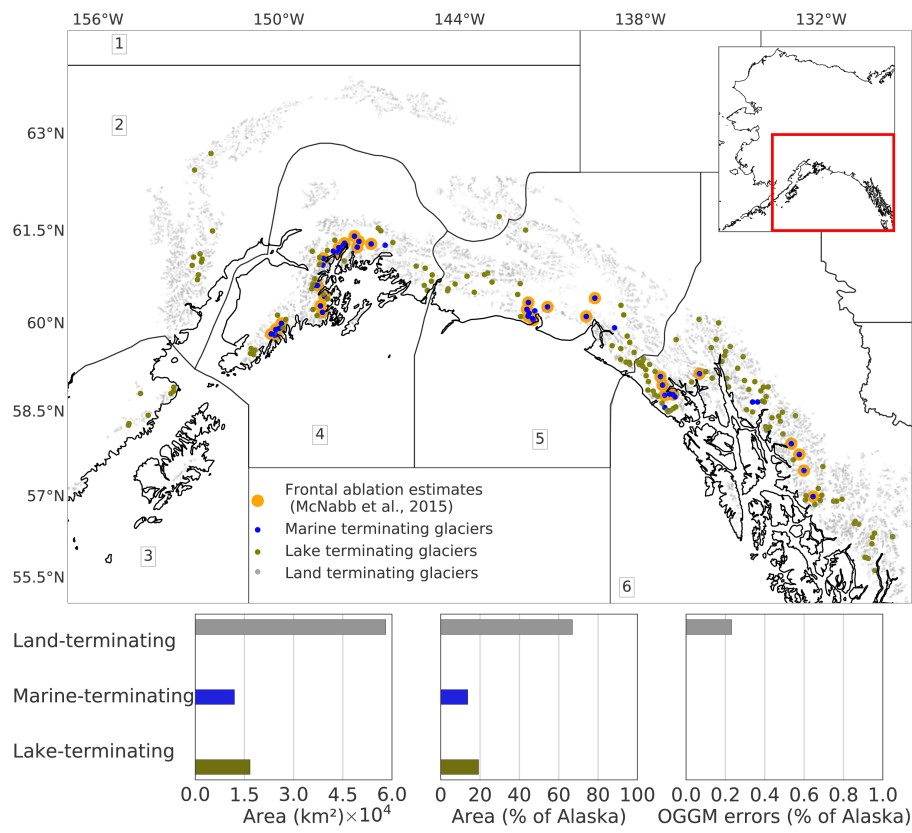

**Figure 1. Upper panel:** map of the RGI sub-regions of Alaska; the dots indicate the location of glaciers classified as land- (grey dots), lake- (olive dots) and marine- (blue dots) terminating in the Randolph Glacier Inventory (RGI v6). Yellow dots indicate the location of the glaciers from which there are frontal ablation estimates (McNabb et al., 2015). **Lower panel:** regional glacier types and basic statistics of the database (area of glaciers per terminus type, regional contribution to the Alaska area in percent, and percentage of the regional area which cannot be modelled by OGGM).

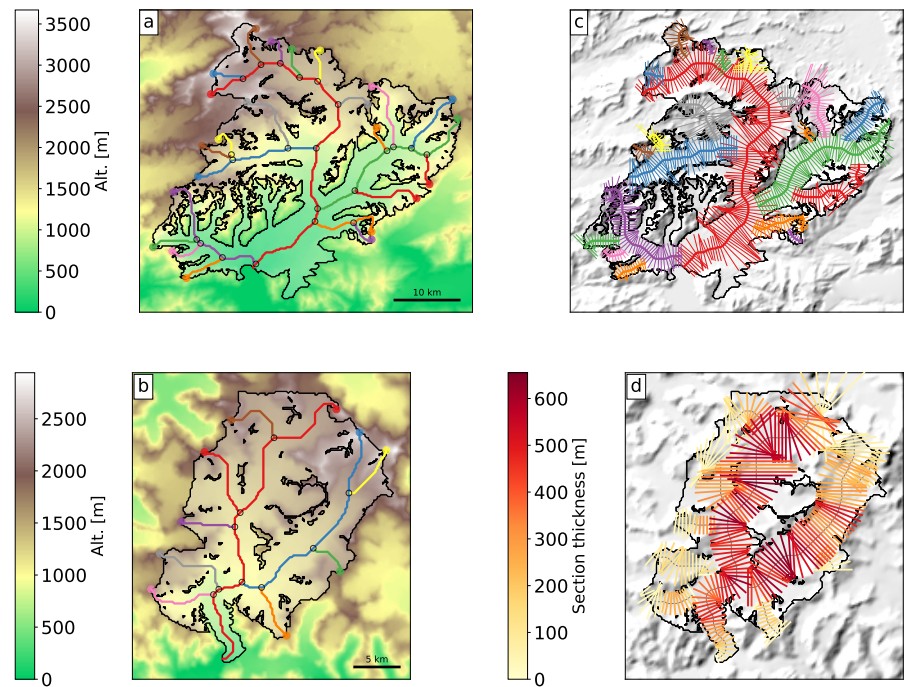

**Figure 2.** Columbia and LeConte Glacier model workflow; **a and b:** topographical data preprocessing and computation of the flowlines; **c:** width correction according to catchment areas and altitude-area distribution; **d:** thickness distribution before accounting for frontal ablation.

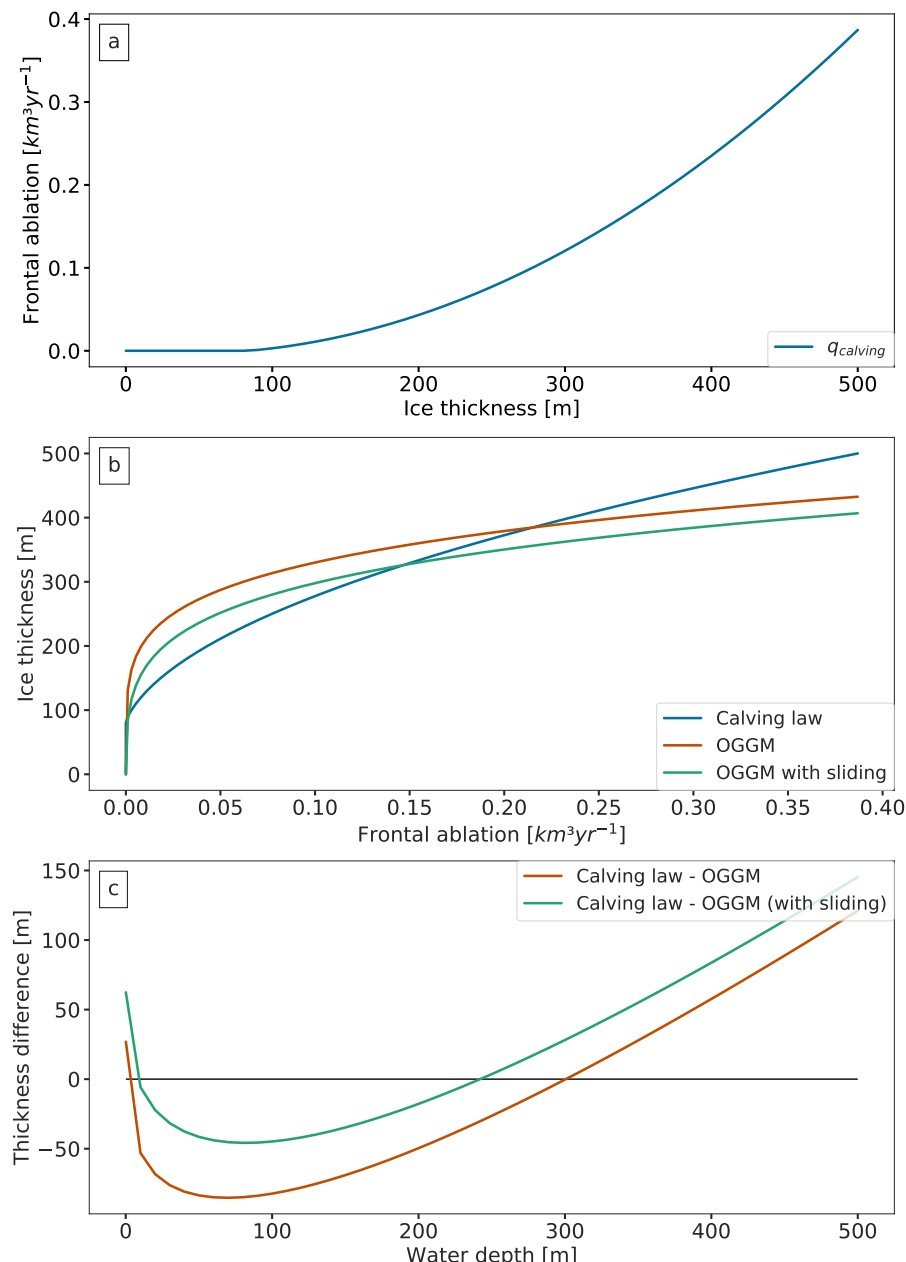

**Figure 3.** Idealised experiments applied to the LeConte Glacier. **a:** Frontal ablation flux computed by the calving law when prescribing a terminus thickness, with $h_f$ ranging from 0 to 500 m. **b:** terminus ice thickness per frontal ablation flux obtained; i) by the calving law (blue curve, same as **a**), ii) by OGGM using ice deformation (orange curve) and iii) by OGGM using ice deformation and adding a sliding velocity (green curve). **c:** illustration of the ice thickness function from Eq. 10 for a given range of water depth values; i) without a sliding velocity (orange curve), ii) with a sliding velocity (green curve).

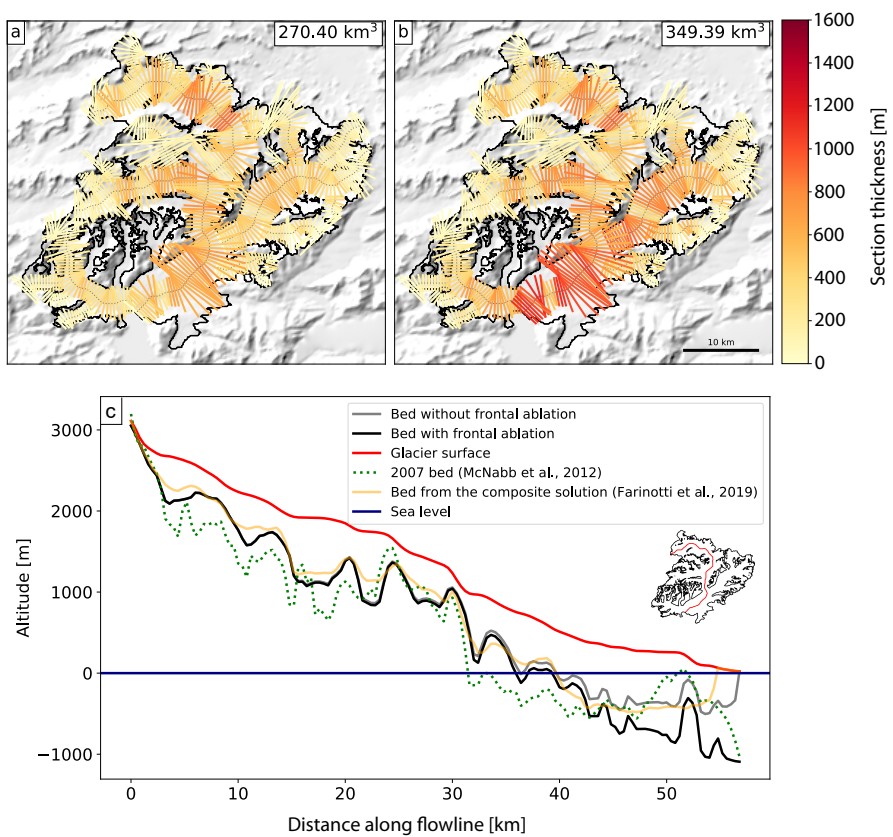

**Figure 4.** Ice thickness inversion results for the Columbia Glacier; **a:** thickness distribution before accounting for frontal ablation; **b:** thickness distribution after accounting for frontal ablation, with a frontal ablation flux computed by the model of 2.98 $km^3$ $y^{-1}$; **c:** Columbia Glacier main centreline profile, comparison between the 2007 estimated bed map (green doted line) from McNabb et al. (2012), the consensus estimate from Farinotti et al. (2019) (orange line) and model output before accounting for frontal ablation (grey line) and after accounting for frontal ablation (black line).

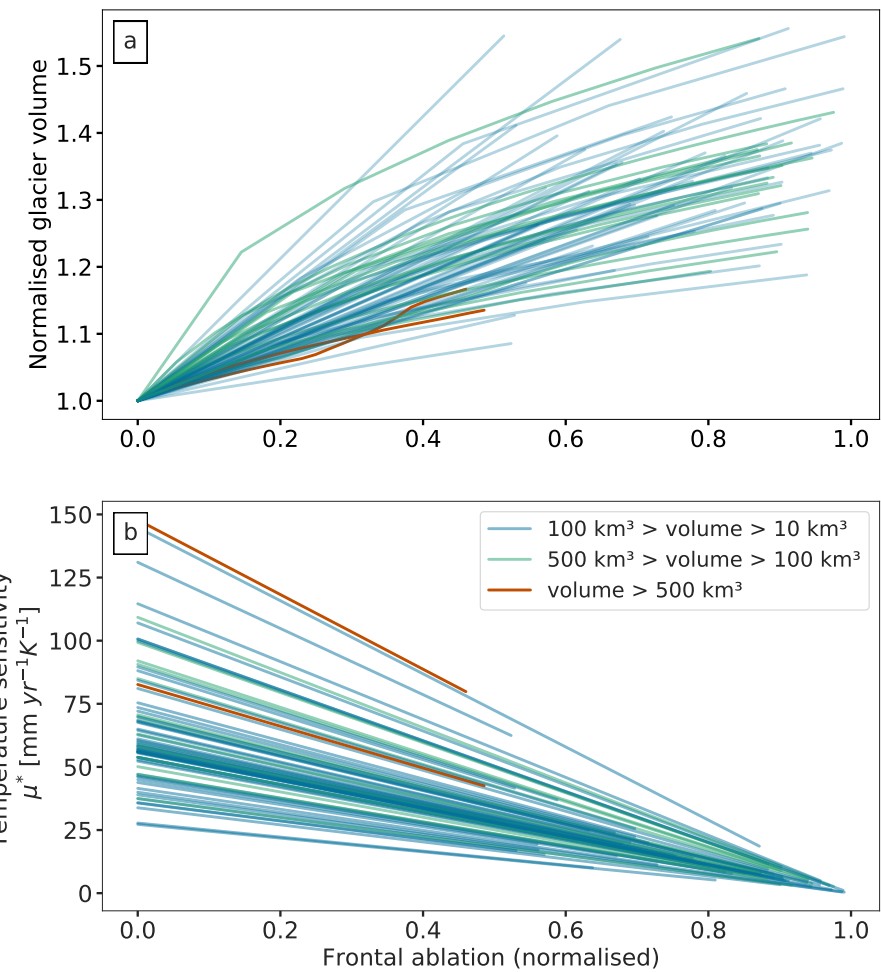

**Figure 5. a:** Normalised glacier volume and **b:** temperature sensitivity ($\mu^*$) of individual glaciers, as a function of the prescribed frontal ablation fluxes normalised by the total accumulation over each glacier. The different colors represents different glacier classes.

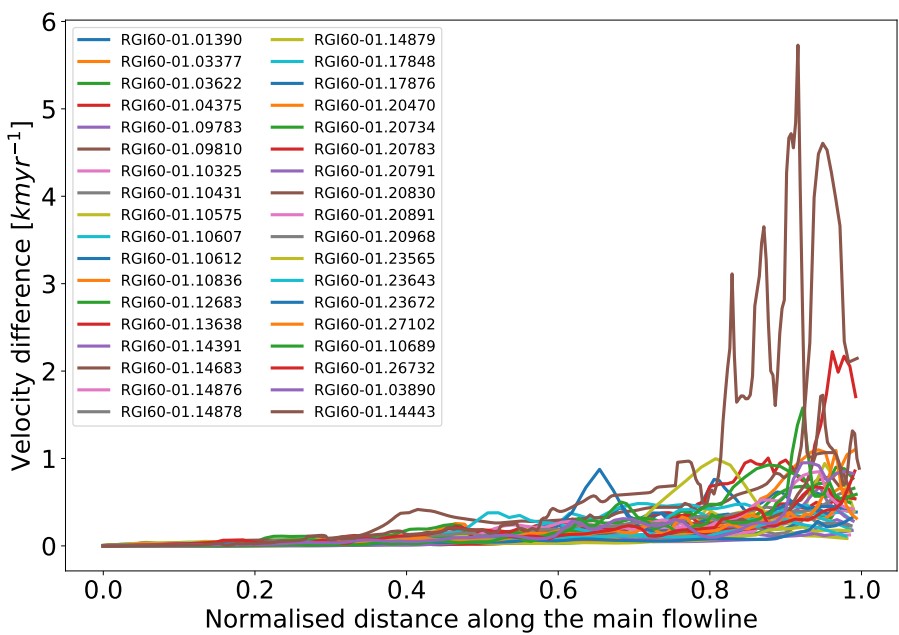

**Figure 6.** Glacier average velocity differences between the two outputs of the model for a subset of marine-terminating glaciers. The differences are between the model output before accounting for frontal ablation and after accounting for frontal ablation in points along the main flowline. The x-axis has been normalised.

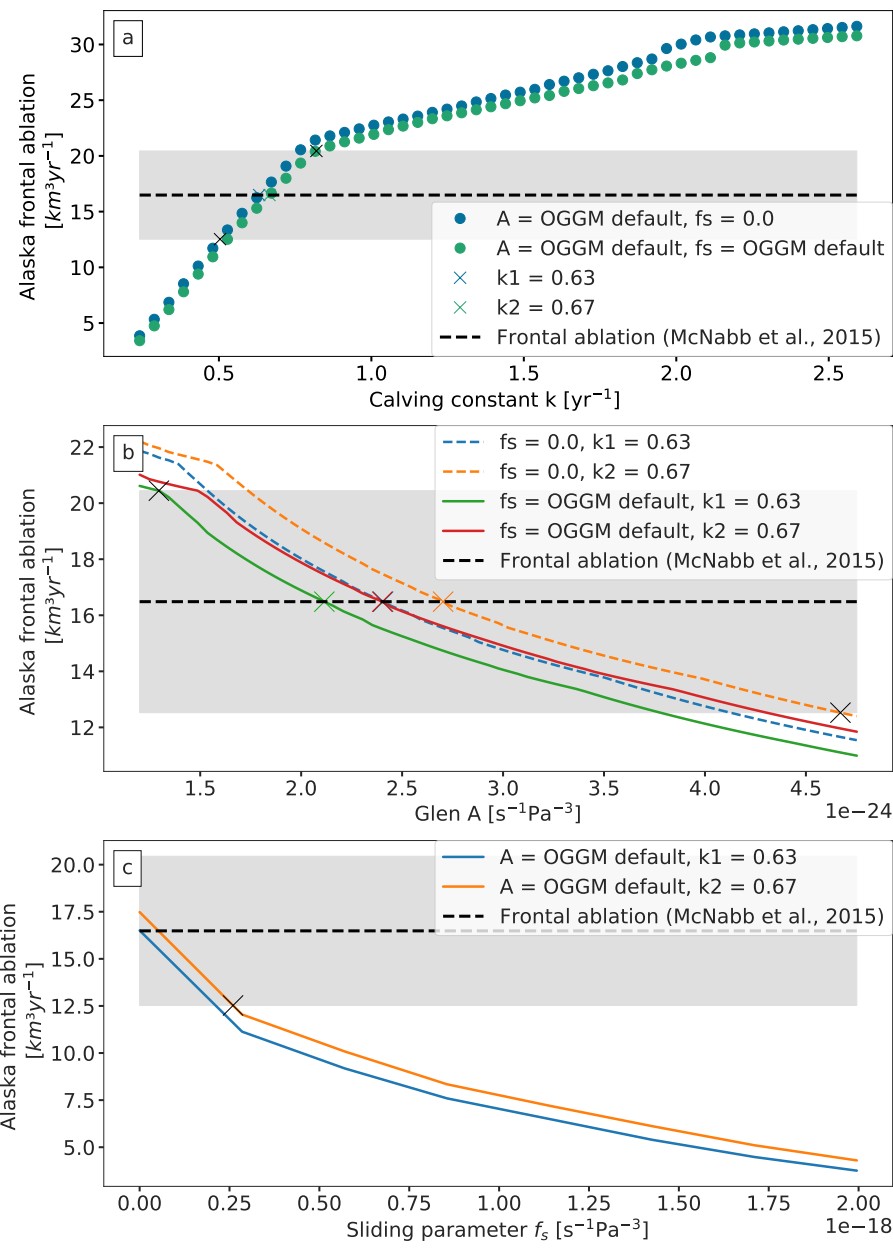

**Figure 7.** Total frontal ablation of Alaska marine-terminating glaciers computed with varying OGGM parameters. The dashed dark line indicates the Alaska regional frontal ablation calculated by McNabb et al. (2015), light gray shading indicating the standard errors as provided in the study. **a:** sensitivity on calving constant of proportionality (*k*); **b:** sensitivity on Glen's A parameter, the coloured dashed lines represent zero sliding; **c:** sensitivity on sliding parameter ($f_s$). Crosses in all plots represent the intercepts between OGGM frontal ablation estimates and McNabb et al. (2015). Note the different y-axis ranges.

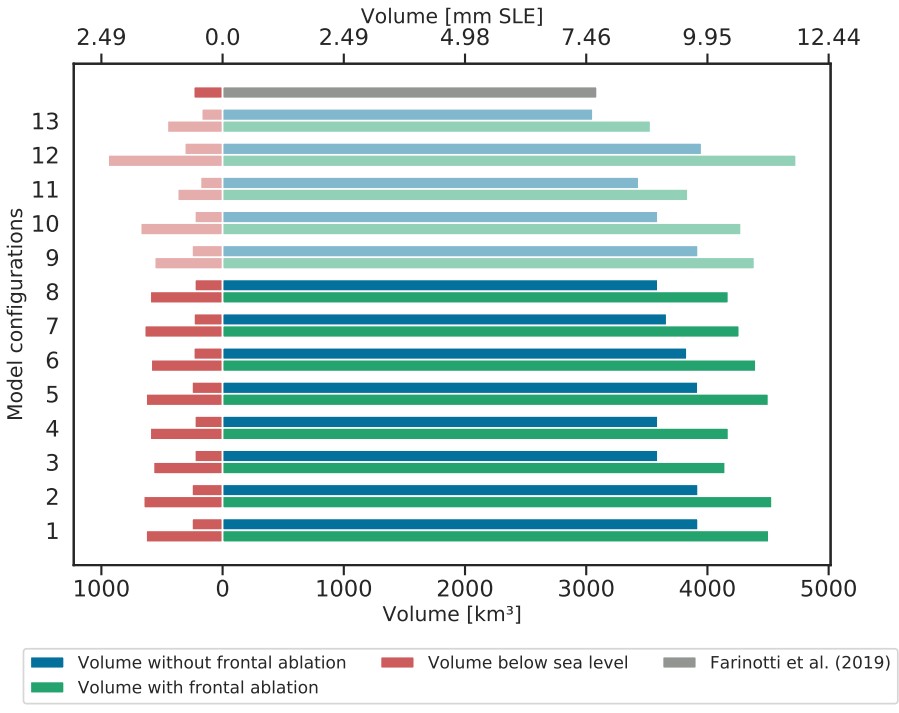

**Figure 8.** Total volume of Alaskan marine-terminating glaciers before (blue) and after (green) accounting for frontal ablation, and the total volume below sea level (red) before and after accounting for frontal ablation. The light shading color bars represent configurations obtained from finding the intercepts between OGGM frontal ablation estimates and the lower and upper error provided by McNabb et al. (2015). The grey bar represent the consensus estimate for these glaciers obtained by Farinotti et al. (2019). The descriptions for each configuration can be found in Table 1.

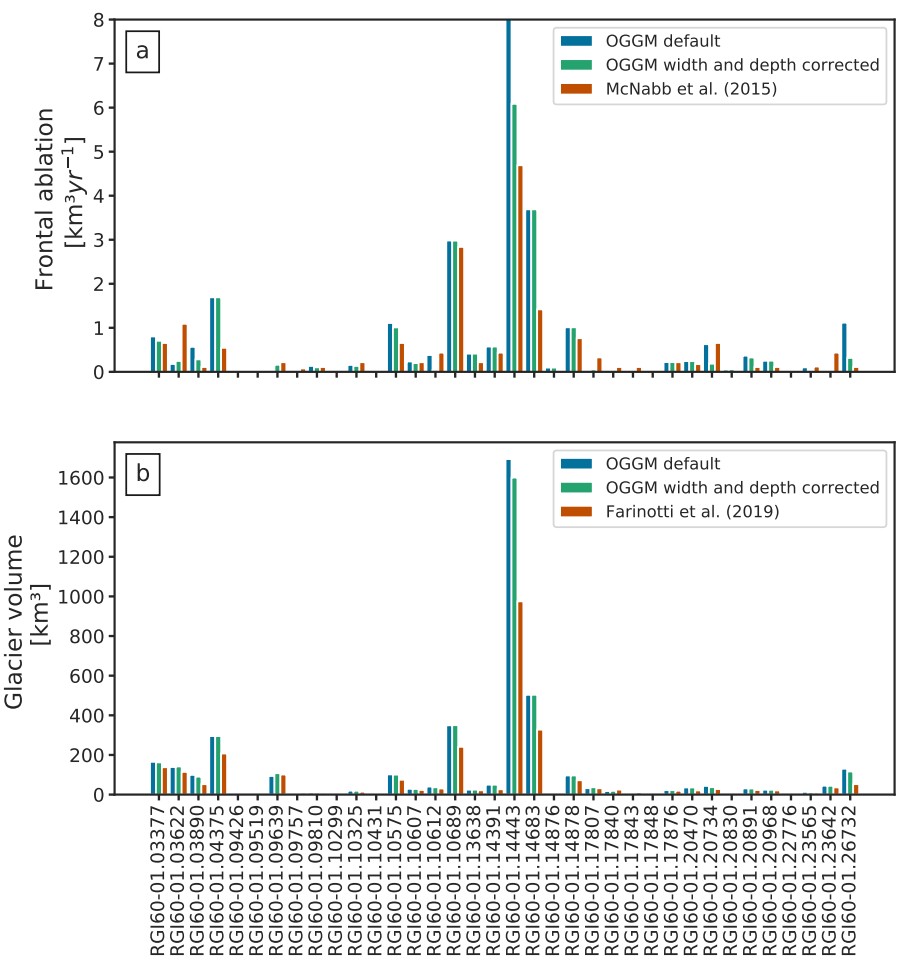

**Figure 9.** Comparison of OGGM (blue and green) $q_{calving}$ and volume estimates for 36 glaciers to **a:** frontal ablation estimates computed by McNabb et al. (2015) and **b:** volume estimates from Farinotti et al. (2019) (red bars). Both OGGM estimates were calculated using default values of $k$, $A$ and $f_s$ (blue) and correcting the width and water depth at the calving front (green). Note that in a, the Hubbard Glacier (RGI60-01.14443) is off scales if no correction is applied to the width and depth of the calving front (blue).

**Table 1.** Different model configurations applied to marine-terminating glaciers of Alaska.

| Experiment number | Calving constant $k$ [yr$^{-1}$] | Glen A creep parameter [s$^{-1}$ Pa$^{-3}$] | Sliding parameter $f_s$ [s$^{-1}$ Pa$^{-3}$] |
|---|---|---|---|
| 1 | 0.63 | default | no sliding, $f_s = 0.0$ |
| 2 | 0.67 | default | no sliding $f_s = 0.0$ |
| 3 | 0.63 | default | default |
| 4 | 0.67 | default | default |
| 5 | 0.63 | $2.41 \times 10^{-24}$ | no sliding, $f_s = 0.0$ |
| 6 | 0.67 | $2.70 \times 10^{-24}$ | no sliding, $f_s = 0.0$ |
| 7 | 0.63 | $2.11 \times 10^{-24}$ | default |
| 8 | 0.67 | $2.40 \times 10^{-24}$ | default |
| 9 | 0.50 | default | no sliding, $f_s = 0.0$ |
| 10 | 0.82 | default | default |
| 11 | 0.67 | $4.67 \times 10^{-24}$ | no sliding, $f_s = 0.0$ |
| 12 | 0.63 | $1.29 \times 10^{-24}$ | default |
| 13 | 0.67 | default | $2.59 \times 10^{-19}$ |

OGGM default values for Glen A = $2.4 \times 10^{-24}$ s$^{-1}$ Pa$^{-3}$ and $f_s = 5.7 \times 10^{-20}$ s$^{-1}$ Pa$^{-3}$. The experiments below the line represent configurations obtained from finding the intercepts between OGGM frontal ablation estimates and the lower and upper error provided by McNabb et al. (2015).