# Peer review of "Impact of frontal ablation on the ice thickness estimation of marine-terminating glaciers in Alaska"

_The Cryosphere, 2018_

## Referee Comment (RC1) · Anonymous Referee #1 · 7 Jan 2019

Summary and comments on the manuscript entitled

**Impact of frontal ablation on the ice thickness estimation of marine-terminating glaciers in Alaska**

presented on 10.12.2018

by

B. Recinos et al.

**Summary**

In this manuscript, the authors forward a refinement of the thickness-estimation module of the Open Global Glacier Model (OGGM v.1.0.1.). The refinement concerns the difficulty to close the overall mass budget of glaciers if they show a calving front. In these cases, frontal ablation has to be considered. Yet this quantity is typically badly constrained because it depends on many oceanographic and atmospheric conditions as well as on the unknown frontal ice thickness, being the target quantity of the module. As a solution, the authors present an iterative procedure that can dynamically infer frontal ice thickness by adapting a melt sensitivity parameter. The method is then applied to all marine-terminating glaciers in Alaska. There, the iterative procedure is calibrated to reproduce observation-based estimates of frontal ablations acquired for many of the prominent Alaskan outlet glaciers. The ice volume estimate is put into perspective by a sensitivity analysis varying several model parameters. The refinement suggests an upward correction of the ice volume stored in the marine terminating glaciers of Alaska, from 9.0 to 10.4 mm sea level equivalent.

The study is well written and clearly motivated. Initially, I was very enthusiastic about the methodology and the results. Therefore, it saddens my heart to report that I might have identified a severe issue in the central iterative procedure. The issue boils down to an under-determination of the mass budget problem when frontal ablation is introduced as a free parameter. Consequently, I have many questions on the performance, stability and convergence behaviour of the presented approach. The answers will certainly require an additional section. Moreover, the sequence in which results are presented gives rise to confusion. I therefore suggest some re-organisation. On this basis, I have to recommend that the manuscript undergoes a major revision and I leave it to the discretion of the editor if he wants to continue to consider this submission for publication in *The Cryosphere*.

**General comments**

**Convergence of iterative procedure**
Looking at your iterative calibration procedure from the perspective of an optimisation, I wonder what target quantity is minimised. In other words, what is the reason for this procedure to converge or reach the stopping criterion. Starting from an initial thickness guess, you infer $F_{\mathrm{calv}}$ and update the temperature sensitivity $\mu^*$ in the surface mass balance equation (5). Then, you re-run the reconstruction and get an updated frontal thickness value. From all involved equations, I cannot see a good reason why the following updates should produce values with a gradually reducing relative differences. A reason for non-divergence is that the thickness update involves a polynomial relation with an exponent smaller than one. Yet even if convergence is reached, I wonder about the physical meaning of this specific solution. Please do not misunderstand me here, but I really think that this is an important point with serious implications for the expedience of your approach. I unfortunately do not have a good suggestion for a useful target quantity or another potential quick fix.

To convince the reader about the functionality of this calibration procedure, I think you have to expand the article by another section, which will elaborate on the stability and the convergence behaviour for a few test cases. I am particularly interested in figures showing the iterative changes in the frontal thickness. Is it monotonous or are the over-shootings. The latter seems unlikely considering the underlying equations. An interesting test would be to check what happens if you started from a too large thickness value (for a well-studied glacier). I would expect an even higher calving flux and thus a further increase in ice thickness. I ultimately miss a relation which counter-balances a steady increase during the iterations. In general, you should assure that the final thickness profile does not depend on the initial thickness guess. Another informative analysis would be to see what happens if the stopping criterion is ignored and you continue the iterations for 100 or even 1000 steps. Do the relative differences in the frontal thickness decrease further? This would be a requirement for the introduction of the suggested stopping criterion.

To put my whole concern in simple words: by introducing a calving flux in the mass budget, you have to reduce the amount of necessary melt (for a balanced situation) This reduction further increase the necessary calving flux in each iteration. To break this run-away cycle, you need another physically motivated relation that penalises either low melt values, high calving fluxes or high frontal thicknesses. Such a counter-balance effect might already be at work by the underlying functional dependencies but without a clear physical motivation.

**Manuscript structure**
The structure of the manuscript is not very clear and only after reading all of the results, I finally got my head around the overall strategy to set up the method. A

major drawback is that the calibration of the proportionality factor $k$ in the calving relation with respect to available regional estimates of the frontal ablation is presented rather late in the text. I think that a calibration section will be very useful at the end of the methodology (P9). This section can also serve to explain that you will use two variants of the model: one with sliding and another without.

**Specific comments**

**Suggestions for the iterative calibration procedure**

**A** *Initial thickness*
Concerning the first two steps in your iterative process (P8L19-22), you determine an initial guess for the calving flux, by assuming a frontal ice thickness which is 1m higher than the surface elevation. I think that it will be beneficial to use the flotation criterion here, making an assumption on the ocean water density. This criterion is simple to implement, it will give a larger first guess and it will therefore speed up your convergence.

**B** *A-priori limits*
The flotation criterion for the frontal ice thickness also provides a lower bound $H_{min}$ to the 'real' frontal thickness value. The reason is that most tidewater glacier will likely be thicker and firmly grounded. An upper bound for the frontal thickness ($H_{max}$) is given by integrating the accumulation field. This will provide the maximum ice flux possible along the glacier. Alternatively, you could integrate the SMB above the ELA. This will give smaller maximum flux values (these will however dependent on $\mu^*$). The maximum flux can then be translated into an upper bound for the frontal thickness value ($H_{max}$) via Eqs. (8-10). A conflict between the upper and lower bound ($H_{min} > H_{max}$), will indicate inconsistencies in the climatology and thereby give useful information.

**C** *Stopping criterion*
The stopping criterion is chosen to be an absolute flux value. In this way, the stopping criterion is easier to be reached for small glaciers with overall lower flux values. I do not think that this is a desirable behaviour and it was not communicated as a deliberate decision. I would therefore suggest that you define the threshold as a fraction of the annually received precipitation volume. If this should not be feasible, you could use a constant values that scales either with glacier area or the terminus width.

**Technical corrections**

At this stage, I refrain from providing a list of detailed comments.

---

## Referee Comment (RC2) · Brinkerhoff (Referee) · 7 Jan 2019

In 'Impact of frontal ablation on the ice thickness estimation of marine-terminating glaciers in Alaska,' the authors extend the thickness estimate technique of Farinotti by allowing for a non-zero terminus flux, which is to say that the modeled glacier may lose mass not only by surface mass balance processes, but also by calving and terminal melt. The authors claim show that failing to include this mechanism of mass loss leads to an underestimation of total glacier volume.

Unfortunately, I think that the paper exhibits methodological inconsistencies which preclude me from making a judgement regarding the veracity of their results. It could be

that I have simply misunderstood the authors' work and intent. In this case, I would require a more thorough description of the methods, along with specific justification for their use, in order to be able to proceed to reviewing the results. An accounting of these issues is as follows:

**Eq. 2** This equation is not valid for non-rectangular cross-sections. Rather, it is depth-averaged velocity for a particular location over a cross-section. To make this into depth and width averaged velocity, we need to introduce a parameterization of $h$ (parabolic, for example), and then width integrate. If we do this (assuming a centerline depth of $h_0$ and margin thickness of zero, we get an additional multi-plicative factor of $\frac{128}{315}$ (assuming $n = 3$). Thus, fluxes are being overestimated by a factor of nearly 3.

**Eq. 6** The interpretation of these symbols doesn't make sense. $\Omega$, in this case needs to be the *contributing area* for a given cross section, not the cross-section itself. This correct form leads to units of kg s$^{-1}$. However, the definition of $F_{calving}$ is in units of volume per time, and thus we have a misfit. This would be (numericall) fine if this parameter were solved for because this error could be absorbed into $k$. However, the authors set this to a value previously computed by Oerlemans and Nick, and thus the scaling of the terminal versus surface fluxes is incorrect.

**Eq. 10** This expression for depth makes no sense to me, partially because the terms included are not well defined. What is the 'elevation of the glacier terminus', $E_t$? We're dealing with vertical ice cliffs here, so is this the base of the cliff (i.e. bedrock elevation) or the top? In either case, the resulting $d$ is not consistent with the definition of depth used in Oerlemans and Nick frontal ablation parameteriza-tion. Also, I fail to understand the difficulty implied about lake terminating glaciers. The definitions are fairly simple: $H_f$ needs to be the terminus ice thickness, $d$ needs to be the water depth. Neither depend on sea level being zero. (This is not to say that there is no difference between marine and lake-terminating glaciers;

[Figure]

$k$ should be different between them).

**Sec 3.4** It is not clear what this iterative procedure accomplishes, especially if $\mu^*$ is being altered, as is indicated. It seems that for a fixed surface mass balance and terminus position, there are any number of valid solutions that respect the constraints that $H_f \geq 0$. Is it trying to match a specific $F_{calving}$ based on observations? In that case, I can see the utility in changing $\mu^*$. But it seems to me that altering $k$ would be more reasonable, since frontal ablation parameterizations are far more uncertain that surface mass balance parameterizations.

The above issues are problematic individually, but taken together, they call into serious question the validity of the results. I forego further comment until such a time as they are addressed.

---

## Author Comment (AC1) · 17 Jan 2019

We would like to thank Douglas Brinkerhoff for taking the time to read our manuscript and give us a chance to answer to his concerns before the end of the review process. We hope that our response is clarifying the motivation behind our study, and we remain available for further questions.

Here we present a detailed point by point response (the reviewer's comments are given in italics, our answer in normal font).

RC: *Eq. 2. This equation is not valid for non-rectangular cross-sections. Rather, it is*

*depth- averaged velocity for a particular location over a cross-section. To make this into depth and width averaged velocity, we need to introduce a parameterization of h (parabolic, for example), and then width integrate. If we do this (assuming a centerline depth of $h_0$ and margin thickness of zero, we get an additional multiplicative factor of $\frac{128}{315}$ (assuming $n = 3$). Thus, fluxes are being overestimated by a factor of nearly 3.*

AR: yes, we should have been more precise in the formulation here. In OGGM we support rectangular, parabolic and trapezoidal bed shapes. For all cases, we compute the ice velocity at the maximum thickness $h_0$, then multiply this velocity by the cross-section area to compute the flux. The section area is $S = \frac{2}{3}h_0w$ in the parabolic and $S = h_0w$ in the rectangular case (with $w$ the section width). This is physically not correct (we are missing a non-linear term in the variation of $\tau$ with the parabola), and is indeed an overestimation of the flux with respect to a true parabolic bed.

Nye (1957) gives analytical solutions as to how much this overestimation could be (his Tables IIIa and IIIb). He gives solutions for the section average velocity and the surface velocity at the parabola's center, i.e. providing an upper bound of the error in our approach: this overestimation ranges from a factor 1.48 to 2.25 depending on the parabola's width to depth ratio.

That being said, we have to emphasize here that OGGM is of the "paremeterized" ice model type, aiming at the simulation of a very large number of glaciers with unknown boundary conditions. We could implement the correct solution (and will happily add an option to use it, as we've recently done by implementing lateral drag shape factors following Adhikari and Marshall, 2012, follow **this link** for the implementation).

However, the current implementation has several practical advantages: it allows us to use the same numerical solver for all bed-shapes (Maussion et al., 2018), is computationally efficient and consistent between the inversion and the forward model. This alone should not be used as an argument to justify our method. More importantly, this simple approach reduces the difference in fluxes between different bed shapes, which

are unknown a priori (and are neither parabolic nor rectangular in reality). Hence, the sensitivity of the model to this unknown parameter is reduced as well.

In the case of the ice thickness inversion, the flux is prescribed by the apparent mass-balance anyway, so that the implied uncertainties are transferred to the cross-section's thickness and volume (and each section is independent from another). The same equations are used to solve for $h_0$ with a given flux. This results in a polynomial of degree 5 with a unique solution for $h_0$ in $\mathbb{R}_+$. Therefore, for any given glacier with unknown bed-shape and prescribed apparent mass-balance, we will have a volume ratio of approx. $\frac{2}{3}$ between the two cases[1]. In practice, OGGM relies on geometrical considerations to decide if the shape is parabolic or rectangular (Maussion et al., 2018).

Finally, it must be added that the inversion model used in this study is the same as the one used in Farinotti et al. (2017) (where OGGM ranked amongst the best models able to process a large number of glaciers), Maussion et al. (2018), and Farinotti et al. (accepted). While this doesn't mean that the model is error-free (there is no such thing anyway), a systematic error of a very large magnitude is unlikely to have been left unnoticed.

The interested reader can find the corresponding code implementation at the following locations (links):

- **ice thickness inversion**

- **forward model**

- **tests of the forward model** where we compare our implementation to the solver by Jarosch et al., 2013.

- **tests of the inversion model in ideal cases simulated by the forward model**
* * *
[1] $\frac{2}{3} \times \frac{3}{2}^{\frac{1}{5}} = 0.72298$ to be exact, in the case $n = 3$ and without sliding

*RC: **Eq. 6**. The interpretation of these symbols doesn't make sense. $\Omega$, in this case needs to be the contributing area for a given cross section, not the cross-section itself. This correct form leads to units of $kgs^{-1}$. However, the definition of $F_{calving}$ is in units of volume per time, and thus we have a misfit. This would be (numerically) fine if this parameter were solved for because this error could be absorbed into $k$. However, the authors set this to a value previously computed by Oerlemans and Nick, and thus the scaling of the terminal versus surface fluxes is incorrect.*

**AR:** you are fully right, $\Omega$ is of course the contributing area (we call it "catchment area" in the model code), this was a bad typo. We convert between volume and mass using an ice density of 900 kg m$^{-3}$. The conversion is implemented **here**.

*RC: **Eq. 10**. This expression for depth makes no sense to me, partially because the terms included are not well defined. What is the 'elevation of the glacier terminus', Et? We're dealing with vertical ice cliffs here, so is this the base of the cliff (i.e. bedrock elevation) or the top? In either case, the resulting d is not consistent with the definition of depth used in Oerlemans and Nick frontal ablation parameterization. Also, I fail to understand the difficulty implied about lake terminating glaciers. The definitions are fairly simple: Hf needs to be the terminus ice thickness, d needs to be the water depth. Neither depend on sea level being zero. (This is not to say that there is no difference between marine and lake-terminating glaciers; k should be different between them).*

**AR:** the water depth $d$ is estimated from free-board, using the terminus elevation obtained by projecting the RGI outline to the DEM. I.e., the terminus elevation is the top of the cliff. The Terminus elevation ($E_t$) minus the thickness of the ice ($H_f$) is the water depth (negative values) at that point, or as Oerlemans and Nick (2005) definition of $d$ (bed elevation with respect to sea level, See Figure 1 from Oerlemans and Nick, 2005). We are not able to estimate a water depth for lake-terminating glaciers, because for that we would need to know the free-board of the glacier terminus, i.e. the elevation of the glacier lake surface (for the elevation of the ocean surface, we assume that it is 0 m a.s.l.).

[Figure]

We leave the discussion about $k$ for the next section.

*RC: **Sect 3.4** It is not clear what this iterative procedure accomplishes, especially if $\mu^*$ is being altered, as is indicated. It seems that for a fixed surface mass balance and terminus position, there are any number of valid solutions that respect the constraints that $H_f \geq 0H$. Is it trying to match a specific $F_{calving}$ based on observations? In that case, I can see the utility in changing $\mu^*$.*

**AR:** Thanks for this comment - we will have to better explain our intent in the manuscript, as this is also a point that Reviewer #1 was asking to improve. We will make further experiments and graphics to explain the procedure, but to expedite the review process, we attempt an explanation here:

First of all, all inversion methods based on mass-conservation assume a mass-flux of zero at the terminus, unless constrained either by a prescribed calving value (e.g. Huss et al., 2015) or by observed ice velocities (e.g. Fürst et al., 2017). In OGGM, we use the equilibrium assumption to calibrate a first guess $\mu$, which assumes an ice flux of zero at the terminus as well. Therefore, if $k$ and water depth (or the calving flux) would be known, we could derive $\mu$ from them.

In the absence of observations, we go the other way around. The main objective of the iteration is to find a frontal ablation flux and ice thickness compatible both with the frontal ablation parameterization and with mass conservation, in order to compute the first ice thickness inversion and initialize the glacier in the model (black line of Figure 3c). This allows calving glaciers in OGGM to have a terminus thickness bigger than zero and a non-zero ice flux at the end of the glacier (Figure 3c and Figure 5), which is then fed again to the mass-balance model to re-calibrate $\mu$.

Importantly, the experiments described in Sect. 4.1, 4.2 and 4.3 use OGGM's default configuration set up (described in Sect 3.1 and 3.3). During these experiments, we do not make use of any observations[2] to restrict or calibrate $F_{calving}$. One of the most im-
* * *
[2]In the second part of the experiments (Sect. 4.4 - 5) is when we make use of previously calculated frontal

portant result of our study (the ratio of underestimated ice volume because of calving) is robust to the parameter choice and remains approximately constant regardless of the true total ice volume.

Physically, $F_{calving}$ in these experiments (and during the iteration) is limited by the amount of annual accumulation. $\mu^*$ changes at each step during the iteration in order to reconciliate the (now non-zero) frontal ablation flux with mass conservation, while the boundary conditions (most importantly, accumulation) remain fixed. To illustrate how $\mu^*$ varies, we refer you to Fig. 4b and Eq. 5, as well as further experiments that will be realized as a result of the questions raised by Reviewer #1.

*RC: But it seems to me that altering $k$ would be more reasonable, since frontal ablation parameterizations are far more uncertain that surface mass balance parameterizations.*

**AR:** Indeed it will be easier to modify the value of $k$ to match previously calculated regional estimates, or individual calving flux observations. But the ultimate goal of OGGM and our iteration method is to compute a frontal ablation flux for any calving glacier of the world (we are not there yet). Many areas do not have regional estimates or with enough individual calving flux observations to constraint $k$.

We will add an extra section and figure to the revised manuscript in order to illustrate the importance of this iterative procedure when we lack frontal ablation observations or additional data (e.g., depth and terminus width) to restrict our boundary conditions. We will also add some idealized experiments to help the readers to understand this section.

*RC: The above issues are problematic individually, but taken together, they call into serious question the validity of the results. I forego further comment until such a time as they are addressed.*

Thank you for your comments. We hope that we were able to restore your confidence
* * *
ablation fluxes from McNabb et al, 2015.

in our results and we will do our best to clarify those points that caused confusion in our manuscript.

**References:**

Adhikari, S. and Marshall, S., 2012, Parameterization of lateral drag in flowline models of glacier dynamics, J. Glaciol., 58, 212, 1119–1132,
doi: 10.3189/2012JoG12J018

Farinotti, D., Brinkerhoff, D., Clarke, G. K. C., Fürst, J. J., Frey, H., Gantayat, P., ...Andreassen, L. M. (2017). How accurate are estimates of glacier ice thickness? Results from ITMIX, the Ice Thickness Models Intercomparison eXperiment. The Cryosphere, https://doi.org/10.5194/tc-2016-250

Farinotti, F., Huss, M., Fürst, J.J., Landmann, J., Machguth, H., Maussion, F., and Pandit, A: A consensus estimate for the ice thickness distribution of all glaciers on Earth, accepted for publication in Nature Geoscience

Fürst, J. J., Gillet-Chaulet, F., Benham, T. J., Dowdeswell, J. A., Grabiec, M., Navarro, F., ...Braun, M. (2017). Application of a two-step approach for mapping ice thickness to various glacier types on Svalbard. The Cryosphere, 11(5), 2003–2032, https://doi.org/10.5194/tc-11-2003-2017

Huss, M., and Hock, R. (2015). A new model for global glacier change and sea-level rise. Frontiers in Earth Science, 3(September), 1–22, https://doi.org/10.3389/feart.2015.00054

Nye, J. F. (1965). The Flow of a Glacier in a Channel of Rectangular, Elliptic or Parabolic Cross-Section. Journal of Glaciology, 5(41), 661–690. https://doi.org/10.3189/S0022143000018670

McNabb, R. W., Hock, R., and Huss, M.: Variations in Alaska tidewater glacier frontal ablation, 1985-2013, Journal of Geophysical Research: Earth Surface, 120, 120–136, https://doi.org/10.1002/2014JF003276, 2015.

Maussion, F., Butenko, A., Eis, J., Fourteau, K., Jarosch, A. H., Landmann, J., Oesterle, F., Recinos, B., Rothenpieler, T., Vlug, A., Wild, C. T., and Marzeion, B.: The Open Global Glacier Model (OGGM) v1.0, Geosci. Model Dev. Discuss., https://doi.org/10.5194/gmd-2018-9, in review, 2018.

Oerlemans, J. and Nick, F.: A minimal model of a tidewater glacier, Annals of Glaciology, 42, 1–6, https://doi.org/10.3189/172756405781813023, 2005.

---

## Author Comment (AC2) · 28 Feb 2019

**Reply to Anonymous Referee # 1 comments for the manuscript: Impact of frontal ablation on the ice thickness estimation of marine-terminating glaciers in Alaska by Beatriz Recinos et al.**

We would like to thank the reviewer for taking the time to read our manuscript and give us insightful comments on the methodology, as these comments led us to formulate new experiments, to improve the code and the manuscript. We agree that this important point was missing from our manuscript and will provide a new section better explaining our method and our objectives.

Here we present a point-by-point response (given in normal font) to the related issues made by the reviewer (given in *italics*). Additionally to this response, we have prepared a **Python Notebook**, where we explain the new experiments and elaborate on the iterative procedure implemented in the Open Global Glacier Model (OGGM) to find a frontal ablation flux. We think that the notebook addresses best the issues raised by the reviewer: our answer is self-contained but is based on these simulations. Interested readers can try the experiments themselves by visiting **the interactive version of the notebook**.

*RC: Convergence of iterative procedure.*
*Looking at your iterative calibration procedure from the perspective of an optimisation, I wonder what target quantity is minimised. In other words, what is the reason for this procedure to converge or reach the stopping criterion. Starting from an initial thickness guess, you infer $F_{calving}$ and update the temperature sensitivity $\mu^*$ in the surface mass balance equation (5). Then, you re-run the reconstruction and get an updated frontal thickness value. From all involved equations, I cannot see a good reason why the following updates should produce values with a gradually reducing relative differences. A reason for non-divergence is that the thickness update involves a polynomial relation with an exponent smaller than one. Yet even if convergence is reached, I wonder about the physical meaning of this specific solution. Please do not misunderstand me here, but I really think that this is an important point with serious implications for the expedience of your approach. I unfortunately do not have a good suggestion for a useful target quantity or another potential quick fix. To convince the reader about the functionality of this calibration procedure, I think you have to expand the article by another section, which will elaborate on the stability and the convergence behaviour for a few test cases. I am particularly interested in figures showing the iterative changes in the frontal thickness. Is it monotonous or are the over-shootings. The latter seems unlikely considering the underlying equations. An interesting test would be to check what happens if you started from a too large thickness value (for a well-studied glacier). I would expect an even higher calving flux and thus a further increase in ice thickness. I ultimately miss a relation which counter-balances a steady increase during the iterations. In general, you should assure that the final thickness profile does not depend on the initial thickness guess. Another informative analysis would be to see what happens if the stopping criterion is ignored and you continue the iterations for 100 or even 1000 steps. Do the relative differences in the frontal thickness decrease further? This would be a requirement for the introduction of the suggested stopping criterion. To put my whole concern in simple words: by introducing a calving flux in the mass budget, you have to reduce the amount of necessary melt (for a balanced situation) This reduction further increase the necessary calving flux in each iteration. To break this run-away cycle, you need another physically motivated relation that penalises either low melt values, high calving fluxes or high frontal thicknesses. Such a counter-balance effect might already be at work by the underlying functional dependencies but without a clear physical motivation.*

AR: To answer this issue we have followed the reviewer's suggestion and are preparing a new

section in our revised manuscript with experiments that illustrate the converges of the parameterisation. This new section will be based on the notebook linked above, where we experiment with the LeConte Glacier (RGI60-01.03622). Note that it is possible to test any another marine-terminating glacier as well (see the **interactive version** of the notebook).

We use a simple calving law borrowed from Oerlemans and Nick (2005), which relates frontal ablation $F_{calving}$ to the frontal ice thickness $H_f$, the water depth $d$ and the terminus width $w$:

$$F_{calving} = kH_f dw$$

with $F_{calving}$ in km$^3$ yr$^{-1}$, $k$ a calibration parameter (default 2.4 yr$^{-1}$) and $d$ the water depth calculated as:

$$d = H_f - E_t$$

where $E_t$ is the free board.

As explained in our manuscript, ice conservation methods applied to tidewater glaciers *must* take into account this mass-flux at the terminus, otherwise the ice thickness is underestimated. In fact, the default OGGM ice thickness inversion procedure assumes an ice flux of zero at the terminus.

In Fig. 1, we examine how this frontal ablation flux would change if we increase the terminus ice thickness, while keeping the free board fixed (the free board is the only variable we know "with certainty", from the DEM surface elevation at the terminus).

[Figure]

Figure 1: Frontal ablation flux computed by the calving law as a function of the terminus ice thickness.

The flux remains equal to zero as long as the frontal ice thickness is not thick enough to reach water, after which the water depth is positive and calving occurs. The calving flux varies with $H_f$ as a polynomial of degree 2.

We are unaware of the *real* value for the calving flux at this glacier. But from here, we can make some very coarse assumptions:

- the Oerlemans and Nick calving law is perfectly exact

- the tuning parameter $k$ is known

- our glacier is in equilibrium (mass-conservation inversion in OGGM)

- ice deformation at the glacier terminus follows Glen's flow law

Under these assumptions, we set up a new experiment where we compute a frontal ablation flux (from the calving law) for a range of prescribed frontal ice thickness (see Fig. 2), then give this flux back to the OGGM inversion model, which will use this flux to compute the frontal ice thickness according to the physics of ice flow (see the **OGGM documentation** or our manuscript for more information). Fig. 2 shows that there is a unique value for the frontal thickness ($H_f$) that complies with **both** the calving law and the ice thickness inversion model of OGGM.

We already know that the calving law relates the ice thickness to the flux with a root of degree two (blue curve of Fig. 2). But for the the orange curve in Fig. 2, it is Glen's flow law, which relates the ice thickness to the flux with a 5th degree root (assuming $n = 3$). Showing the reason why there is one (and only one) non-zero solution to the problem of finding a calving flux; a flux which is compatible with both the calving law and the physics of ice deformation (under our simplified framework).

Note that changing Glen's deformation parameter $A$ or adding sliding does not change the problem: we will still solve a polynomial degree 5 in OGGM, **with a new term in degree 3** (see green curve in Fig. 2).

[Figure]

Figure 2: Ice thickness per frontal ablation flux calculated at each iteration. The orange line is the flux calculated by prescribing a thickness and using the calving law. The blue line is the flux calculated by OGGM using the iterative procedure. The green line is the flux calculated by OGGM using the iterative procedure and adding a sliding velocity.

There are several ways to find this "optimal" calving flux (or optimum frontal ice thickness), where mass-conservation inversion and the calving law are compatible. In OGGM, we implement an iterative procedure converging to this value in a few iterations (see Fig. 3).

The procedure starts with an initial water depth of 1 m (arbitrary choice that might change in the next version of our manuscript), then iteratively feeds the calving flux back to the mass-conservation inversion function of OGGM, which adapts the mass-balance model to cope with a non-zero ice flux at the front. In order to do that, the temperature sensitivity of the glacier $\mu^*$ has to be reduced (per construction, the original $\mu^*$ is defined such that the flux at the front is zero). Convergence is reached when the OGGM flux equals the calving law flux within 0.1%. Thanks to the uniqueness of the solution, the method always converges regardless of the starting water depth (Fig. 4).

[Figure]

Figure 3: Right: Frontal ablation flux at each iteration. b: Temperature sensitivity $\mu^*$ of the glacier at each iteration.

[Figure]

Figure 4: Frontal ablation flux calculated from different starting points. The different colors represent different water depths at the beginning of the iteration procedure.

However, for some glaciers the calving flux given by the calving law is larger than a flux that can be explained by climate alone, i.e. even without melt, the computed flux is larger than the total accumulation over our glacier. This can happen for several reasons:

- precipitation is underestimated

- the flux is overestimated because of uncertainties in k and the terminus geometry

- the equilibrium assumption is not valid

During these conditions our iterative search can "overshoot". Fig. 5 simulates this case where we set an unrealistically large calving parameter ($k$ equal to $10$ $\text{yr}^{-1}$ in this experiment). If this happens during the iteration, OGGM is going to set $\mu^*$ to zero and compute the corresponding flux (the maximal physically possible value).

***RC: Manuscript structure*** *The structure of the manuscript is not very clear and only after reading all of the results, I finally got my head around the overall strategy to set up the method. A major drawback is that the calibration of the proportionality factor k in the calving relation with respect to available regional estimates of the frontal ablation is presented rather late in the text. I think that a calibration section will be very useful at the end of the methodology (P9).*

[Figure]

Figure 5: a: Frontal ablation flux per iteration. b: Temperature sensitivity $\mu^*$ per iteration. With a $k$ value of 10 yr$^{-1}$.

*This section can also serve to explain that you will use two variants of the model: one with sliding and another without.*

AR: We would like to clarify that the calibration of the proportionality factor $k$ with respect to available regional estimates of frontal ablation does not take part in the iteration procedure described in section 3.4. During the iteration procedure and in most of the study experiments, we keep $k$ constant with a value of 2.4 $yr^{-1}$. We only modify $k$ to match regional estimates during the sensitivity experiments (see section 4.4 and 4.5) with the intention of showing that regardless of the parameter configuration chosen, we can still estimate the relative part of the ice volume that is missed when ignoring frontal ablation in the inversion. We consider that this part of the text does not belong on the methods section, since our optimum goal is to make OGGM independent of frontal ablation observations. Hopefully, the clarification of the iterative procedure above helps clarify this point, too.

***RC: Specific comments.***
*Suggestions for the iterative calibration procedure.*
*A. Initial thickness*
*Concerning the first two steps in your iterative process (P8L19-22), you determine an initial guess for the calving flux, by assuming a frontal ice thickness which is 1m higher than the surface elevation. I think that it will be beneficial to use the flotation criterion here, making an assumption on the ocean water density. This criterion is simple to implement, it will give a larger first guess and it will therefore speed up your convergence.*
*B. A-priori limits*
*The flotation criterion for the frontal ice thickness also provides a lower bound Hminto the "real" frontal thickness value. The reason is that most tidewater glacier will likely be thicker and firmly grounded. An upper bound for the frontal thickness (Hmax) is given by integrating the accumulation field. This will provide the maximum ice flux possible along the glacier. Alternatively, you could integrate the SMB above the ELA. This will give smaller maximum flux values (these will however dependent on $\mu^*$). The maximum flux can then be translated into an upper bound for the frontal thickness value (Hmax) via Eqs. (8-10). A conflict between the upper and lower bound (Hmin > Hmax), will indicate inconsistencies in the climatology and thereby give useful information.*
*C. Stopping criterion*
*The stopping criterion is chosen to be an absolute flux value. In this way, the stopping criterion is easier to be reached for small glaciers with overall lower flux values. I do not think that this is a desirable behaviour and it was not communicated as a deliberate decision. I would therefore*

*suggest that you define the threshold as a fraction of the annually received precipitation volume. If this should not be feasible, you could use a constant values that scales either with glacier area or the terminus width.*

AR: For the issues addressed here, please refer to the first section of the reply: convergence of iterative procedure.

In addition to the changes mentioned above, modifications to the code were needed in order to correct a bug found in the iterative procedure which now makes the result of the parameterisation compatible with mass conservation. The changes implemented can be found **here**.

We will add an explanation of all the modifications to the code in our revised manuscript.

Our main result of the ice volume underestimation when ignoring frontal ablation is still the same after this correction.

We thank you for your comments again and for helping us to improve our method. We will do our best to clarify those points that caused confusion in our manuscript.

**Reference:**

Oerlemans, J. and Nick, F.: A minimal model of a tidewater glacier, Annals of Glaciology, 42, 1–6, https://doi.org/10.3189/172756405781813023, 2005.

---

## Referee Report (RR1)

**Reviewer response to 'Impact of frontal ablation on the ice thickness estimation of marine-terminating glaciers in Alaska'**

May 15, 2019

In 'Impact of frontal ablation on the ice thickness estimation of marine-terminating glaciers in Alaska,' the authors extend the thickness estimate technique of Farinotti by allowing for a non-zero terminus flux, which is to say that the modeled glacier may lose mass not only by surface mass balance processes, but also by calving and terminal melt. The authors show that failing to include this mechanism of mass loss leads to an underestimation of total glacier volume. They explore its sensitivity to a variety of parameter choices. They then apply the method to Columbia Glacier, and then the RGI for coastal AK.

Overall, the paper is much improved over its first version, however I remain skeptical on a few points, some new, some remaining from the paper's first submission.

- Regarding my discussion of a missed factor in the handling of width-averaged fluxes.

  This reply isn't entirely satisfactory. At the end of the day, the authors' stated equations are incorrect, and they should be corrected before publication: $u$ is stated in Eq. 2 as the average cross-sectional velocity, while it is clearly not, *so long as the authors allow for their modelled cross-sections to be non-rectangular.* The authors should either acknowledge that their equations are an approximation that could be off by a substantial factor, or they should implement the analytical solution to $u$ as a function of cross-section shape. The latter would be preferable, but the former is also reasonable. However, simply writing an incorrect equation and then making an appeal to antiquity ("A systematic error a very large magnitude is unlikely to have been left unnoticed") isn't defensible.

  The argument that by specifying the flux via integration of the upstream effective mass balance, this makes the errors in the resulting thickness go away is also not correct. Here, we have that $q = uS$, where $u$ is the depth and width average velocity given by

  $$u = \int_W \bar{u}(h(y)) \, \mathrm{d}y = k\bar{u}(h_0),$$

where (in the absence of lateral drag) $k \leq 1$, with $k = 1$ for a rectangular cross section, and $k = \frac{128}{315}$ for a parabolic one. This leads to the equation

$$q = p(kf)wh_0^{n+2},$$

where $p = \frac{2A}{n+2}(\rho g \alpha)^n$, $f$ is the ratio of the cross-sectional area to a rectangular one (i.e. 1 for rectangular, $2/3$ for parabolic), and $w$ is the width. Solving for $h_0$ gives us

$$h_0 = (kf)^{-\frac{1}{5}} \frac{q}{pw}^{\frac{1}{5}}.$$

For the rectangular case, of course the geometric prefactor is unity, but for the parabolic case, it's around 1.3. *Since this factor is so straightforward to calculate, why not just include it?* It requires no separate solver, just the imposition of the correct geometric pre-factor when estimating $h_0$. If the bed shape isn't known (which it usually isn't), then the fact that the pre-factor is unknown should be included in error estimates.

- Regarding the definition of depth.

  I think it would be better to develop the depth definition such that it could be applied to situations in which the water elevation isn't zero, e.g. for lakes. Thus, we would have that

  $$d = H_f - E_t + z_w,$$

  where $z_w$ is the water surface. After such a definition, if there is in fact no way to measure the surface elevation of water bodies that are not the ocean, then one can explicitly set $z_w$ to zero.

- Iterative calibration procedure. I think that the five steps outlined in this section are a bit confusing because some of them are done several times. Would it be possible to put this in pseudocode, as is typically done in math/CS papers? The LaTeX algorithm environment is designed for just such a task.

  Also, after a fair bit of fiddling with equations, I managed to convince myself that this problem does have a unique solution, and that this solution induces a unique value of $\mu^*$. Indeed, the explanation as it appears in section 3.5 is, I think, correct. However, it's a little bit difficult to understand, and should also be placed in Sec. 3.4. I would explain it as such: there are two equations in this model that can be used to estimate the ice flux given the thickness: Glen's flow + sliding law (quintic in thickness), as can the calving law (quadratic in thickness) under the assumption that the amount of ice calved is equal to the amount delivered to the terminus. These fluxes must be equal, which leads to an equation which is quartic in thickness, the unique real-valued solution of which yields the thickness value consistent with both Glen's and calving laws. Inverting either of these laws to get an equation for terminus flux and applying the

computed thickness value leads to the one and only terminal flux that is consistent with both thickness estimation methods. The authors impose this specific flux in OGGM by adjusting the degree day factor $\mu^*$. Interestingly, being quartic, I think that the optimal thickness has an analytical solution: instead of computing it directly, the authors are using a fixed-point iteration to find the roots (which I don't object to at all).

In any case, a clearer description of what exactly this calibration procedure is doing mathematically would help the skeptics amongst us. Also worth noting: instead of adjusting the degree day factor, one could adjust other parameters (i.e. sliding law, calving parameter) such that this relationship is satisfied as well. A little bit better justification of why $\mu^*$ is the one to fiddle with is in order.

- "This can be due to two equally likely factors: solid precipitation is underestimated, or the frontal height and therefore frontal ablation is overestimated". Alternatively, $k$ is overestimated or the calving law is wrong.

- In the Columbia Glacier case study, would it be possible to report the terminal velocities that OGGM predicts, as a way to see whether these are remotely consistent with observations? Columbia is, of course, very fast, and if the velocities are too low, this could lead to an overestimation of near-terminus thickness.

- In Section 4.3, the authors compare the velocities produced by the model to the case where frontal ablation is not considered, which is useful. More useful, I think, would be comparing these velocities to measured ones (e.g. from GoLive) in order to get a sense of whether the physics being used by OGGM are consistent with reality.

- In Section 4.4, the authors perform a sensitivity study over various physical parameters. This is an important step, but I think that the range of possible basal tractions is far too small. Velocities often vary between tidewater termini by two orders of magnitude. I would like to see the basal traction parameter vary over similar ranges, primarily because this parameter aliases differences in effective pressure: alternatively the sliding law used in this paper could include effective pressure explicitly, computed using height above overburden as an upper bound.

- The Discussion isn't really a discussion, it's another methods/results section in which the authors include additional observations in their inversion. I would suggest relabeling it as such.

---

## Referee Report (RR2)

Summary and comments on the manuscript entitled

**Impact of frontal ablation on the ice thickness estimation of marine-terminating glaciers in Alaska**

presented on 10.12.2018

by

B. Recinos et al.

**General comments**

First, I want to express my appreciation for the serious efforts to address the concerns is raised on the convergence. The authors now clearly show that the iterative procedure does converge to the same calving front thickness starting from various initial thickness values.The authors make clear that this iterative procedure is actually nothing else than solving a polynomial of degree 4 in two variants: with and without sliding. This also means that there is an analytical solution to the calving front ice thickness $H_f$ and thus the calving flux $F_{calving}$. This resolves my initial concern on the 'missing' target quantity of the iterative optimisation. It is inherent in their formulation. My sincere apologies that it took me a moment to get my head around this. However, in the presented iterative scheme the $\mu^*$ calibration is entangled with the iterative determination of the calving flux $F_{calving}$. This makes the iterative procedure unnecessarily complex and blurs which variables actually determine $F_{calving}$. In my opinion, it is possible to neatly separate the calving flux problem from the $\mu^*$-calibration. If I am wrong here, please justify well and ignore my comments below. Otherwise, please disentangle these two steps to avoid needless complexity. The presentation of the procedure will strongly profit in terms of clarity.

In essence, your iterative procedure boils down to setting the flux values in Eqs. (1) and (8) to be equal at the calving front. In this way, the calving flux is eliminated and it no longer needs to be determined by integrating the apparent mass balance along the glacier centreline. The solution for $H_f$ is thus independent of $\mu^*$. The resultant polynomial of degree 4 has an analytical solution and you could avoid any iterative procedure. Moreover, an analytical solution would add to the computational efficiency of OGGM. Though I prefer an analytical formulation, I refrain from urging the authors to implement the equations for the root determination. However, I urge the authors to make clear that these iterations are simply a numeric strategy to solve for the real roots of a 4th order polynomial. Clarify that this polynomial is independent of the apparent mass balance integration. You should also consider to simplify and streamline the iterative procedure.

Here, my suggestion:

**Step 1** Determine initial thickness $H_f^0$ guess (1/3 $E_f$ or whatever) at calving front and compute a first guess for the calving flux $F^0_{calving}$ using Eq. 8. Set i=0.

**Step 2** Use $F^i_{calving}$ to update the ice thickness $H_f^{i+1}$ using Eq. (1) either with or without sliding.

**Step 3** Use Eq. 8 to update the calving flux $F^{i+1}_{calving}$.

**Step 4** Set i=i+1. Iterate **Step 2** and **Step 3** until convergence is reached.

I think the above scheme becomes much easier to implement, if you swap Eqs. (8) and (1). Then, you would only need to solve for the roots of a polynomial of degree 2 (Eq.8) to update the thickness guess in **Step 2**. The polynomial of degree 5 (Eq. 1) would then only be evaluated in **Step 1, Step 3** by inserting a the iteratively updated thickness value.

Once the calving flux $F_{calving}$ is determined, you simply add it to the subsequent $\mu^*$-calibration. You could even keep the condition that $\mu^*$ should not become negative. The structure of the manuscript would need to be slightly adjusted to clarify the one-way dependence of $\mu^*$ on $F_{calving}$ and NOT vice versa.

**Specific comments**

- Explain the improvement of using Eq. (8) to prescribe the calving flux for Eq. (1). Why not simply use a fixed value per region as in Huss and Farinotti (2012). What is the physically added value of the parameterisation from Oerlemanns (2005). A motivation of this choice is missing (P7L30ff). Why is this a good or even better choice?

- Rethink the necessity of section 3.5.

- You could check internal consistency: Remember $F_{calving}$ used for the $\mu^*$-calibration. Run the thickness reconstruction and insert $H_f$ into Eq. (8). Do you get the same value?

**Technical corrections**

P7L13ff Comparing Eq. (6) and (7), it seems that q is equal to $F_{calving}$ at the calving front. You could use one variable name for the flux, either $F$ or $q$.

P11L2 What is the default value for $k$. If you used $k = 2.4yr^{-1}$ here, this would mean that after the regional calibration to $k = 0.66yr^{-1}$, the Columbia calving flux would decrease by a factor 4. This is important. In this case, the

McNabb values would no longer compare well and you should reconsider your regional calibration. Please specify the default value for $k$ and discuss.

P11L4  Fig.7 $\longrightarrow$ Fig.6 (please re-check all figure references.)

P13L24  '..., we increase ...' $\longrightarrow$ 'estimate is increased from ... to ...'

---

## Referee Report (RR3)

**Reviewer response to 'Impact of frontal ablation on the ice thickness estimation of marine-terminating glaciers in Alaska'**

**July 31, 2019**

In 'Impact of frontal ablation on the ice thickness estimation of marineterminating glaciers in Alaska,' the authors extend the thickness estimate technique of Farinotti by allowing for a non-zero terminus flux, which is to say that the modeled glacier may lose mass not only by surface mass balance processes, but also by calving and terminal melt. The authors show that failing to include this mechanism of mass loss leads to an underestimation of total glacier volume. They explore its sensitivity to a variety of parameter choices. They then apply the method to Columbia Glacier, and then the RGI for coastal AK.

Overall, the paper is improved from its previous versions. I suggest the following technical corrections

- Returning once again to the discussion of a missed factor in the handling of width-averaged fluxes, the paper needs to be specific in how the omission could affect the results, e.g. "neglecting variations of *u* across the glacier width may lead to an overestimation of fluxes by a factor of xx, however we proceed with this approximation because it allows for easier generalization to alternative bed shapes." The authors have already written something similar in their response, but it does not enter the manuscript.
- I think that Eq. 7 still has a units error.  $\int_{\Omega} \rho \frac{\partial h}{\partial t} dA$  has units of mass per time, yet this is equated to  $q_{calving}$  which has stated units of length per time. Which is it, and is there an equivalent mistake in the code?
- Regarding the definition of depth, I didn't mean to say that you have to include lake-terminating glaciers in the analysis, simply that it is better to include the complete derivation of a method, then simplify it later, which is to say that you can write

$$d = H_f - E_t + z_w,$$

and then say that you are setting  $z_w$  to zero for the rest of the paper. Would it not be valid to use this method if I started measuring water surface from some other arbitrary datum than mean sea level? I think that it would, and by using the above definition, you retain that capacity, even if you don't use it.

- I wrote in my previous review: In the Columbia Glacier case study, would it be possible to report the terminal velocities that OGGM predicts, as a way to see whether these are remotely consistent with observations? Columbia is, of course, very fast, and if the velocities are too low, this could lead to an overestimation of near-terminus thickness. I didn't write this comment out of pure curiosity, but rather because I think it needs to be demonstrated that this model produces sane results for these tidewater glaciers. Thus, I think that the authors' response to this point should be included in the paper somewhere.
- In Section 4.2, paragraph 2, the authors state that smaller glaciers change relatively more than large glaciers for a specified frontal ablation. I think that the different scale of the glaciers overwhelm potential differences in dynamics, making these results less interesting than they could be. To me, it's rather obvious that specifying a given and fixed calving flux would have a larger relative effect on a smaller glacier, because it's a relatively larger portion of the mass budget. A more interesting experiment might be to specify a range of frontal ablation fluxes as fractions of the total accumulation, e.g.  $q_{calving} = f \int_{\Omega} p_f P^{solid} dA$  for  $f \in [0, 0.5]$ . Thus the frontal ablation would naturally scale with the glacier size and we could see the result of the differences in physics between small and large glaciers.

---

## Referee Report (RR4)

Summary and comments on the revised manuscript entitled

**Impact of frontal ablation on the ice thickness estimation of marine-terminating glaciers in Alaska**

first presented on 10.12.2018

by

B. Recinos et al.

**Recommendation**

The authors have put much effort in adjusting the code and revising the manuscript in response to the second review round. I highly appreciate these changes and I am sure that the manuscript did gain in clarity. I am therefore only left with some few last minor comments and suggestions, which the authors should address before the article is considered for final publication.

**Specific comments & suggestions**

P6L14   Check the units of the sliding parameter. I think that they should be $m^2 s^{-1} Pa^{-3}$.

P8L7   Please remove the $k$-value as given by Oerlemanns & Nick (2005). It gives rise for confusion later on where you mention default values (P10L21).

P9L18-19   Please remove the first argument against implementing an analytical solution because the history of the review process will not be visible to a wide audience.

P10L5-10   You should remember the reader about the calibration of the calving factor $k$ in Sect. 2.3. This calibration defines the default values for $k$, $A$ and $f_S$. Give their values or refer to Table 1.

P10L28-32   I think that it is if high interest to further expand on the Columbia Glacier case study. Please also compare your results to the 'consensus estimate' of glacier ice thickness from Farinotti et al. (2019). Is there a qualitative difference? This is not a lot of work. Just add an extra line in Fig. 4c and add 2-3 sentences of explanation. To put this comment in a question: Why should we prefer the OGGM thickness values over the consensus thickness estimate for marine terminating glaciers in Alaska?

P11L30-P12L3   This passage is confusing because it evokes alternative choices in the methodology, which were not pursued. Please consider to transfer it to the methodology or remove it.

P12L29    I understand that you define the equally good parameter sets on the basis of Fig. 7. From Table 1 , I see the following ranges: $A$ from 2.4 via 2.41 to 2.7 $10^{-24} s^{-1} Pa^{-3}$, $k$ varies between 0.63 and 0.67, and $f_S$ is only turned off and on. The creep parameter $A$ is therefore only increased. Are the two values 2.4 and 2.41 different settings? Why did you not decrease this value. The MacNabb frontal ablation uncertainty would allow a total range of roughly 1.5 to $4.0 10^{-24} s^{-1} Pa^{-3}$. The sliding parameter $f_S$ is only decrease to zero, which hardly affects the frontal ablation (as visible from Fig. 7c). Again, the associated uncertainty in frontal ablation would allow a larger range up to $20.0 10^{-18} s^{-1} Pa^{-3}$. Finally, the $k$-range seems very narrow. You only use the two calibrated values for the cases with and without sliding. Again the frontal ablation range from MacNabb et al. (2015) would allow a range from 0.5 to 0.75. In summary, the question is why you chose such a narrow and selective parameter range? Please motivate your choice or adjust.

P12L29-P13L6    I do miss a description of the sensitivity results due to changes in all parameters. You only say that the creep and sliding parameters are most influential. What is the relative effect on regional ice volume. From Fig. 8, I loosely infer ±5%. How does this sensitivity compare to the uncertainty of other reconstruction approaches. The 5% value seems optimistic and it is certainly linked to your choices of the parameter ranges.

P13L15    Most parts of the discussion section focuses on the importance of correct geometric information near the glacier terminus. This is certainly valuable but I think you should use this section to also compare your approach with other reconstruction estimates (e.g. Farinotti et al., 2019). In my view, your results will show a clear improvement. Moreover, you should also discuss the choices you made in the sensitivity analysis.

P14L6    What errors do you refer to here? I think you refer to the width and depth correction that is described thereafter.

P15L14    Remove the parenthesis and reformulate. This information is decisive for the sentence.

Fig. 4c    Could you add the result for Columbia Glacier from Farinotti et al., (2019) in panel (c). It is of great interest to the community how their consensus estimate does perform on tidewater glaciers.

Fig. 5b    The sensitivity of the inferred $\mu^*$ on the calving factor $k$ indicates that $\mu^*$ is set to zero for most of the smaller glacier (¡100km2) even for the default $k$ value. This issue needs discussion in Sect. 5.

---

## Author Response (AR2)

**Point by point response to editor and reviewers - second revision**

We would like to thank you all for your constructive feedback on our manuscript and for your patience during this review process. We have addressed all points raised by the reviewers and hereby submit a new version of the manuscript together with a point by point reply to each of the editor's and reviewers' comments.

We modified the frontal ablation parameterisation tool and were able to eliminate the iteration procedure as suggested by the reviewers. We now find a frontal ablation flux by implementing a Bound-Constrained Minimization Method and solve for the ice thickness equations numerically (See Sect. 3.3 for details). The changes implemented in the code can be found **here.**

These modifications did not change the model results, but provide several advantages: i) the method is independent of the initial conditions (e.g. initial water depth), ii) the solution for the frontal ice thickness is independent of the temperature sensitivity $\mu^*$ (but we kept the condition that $\mu_*$ may not become negative in order to comply with mass conservation), and iii) the method is generalizable to other formulations of ice deformation or calving law (see text and discussion below).

Additionally to this response, we have updated the **Notebook** used in the previous reply, where we explain the changes in the new code and the method's limitations. Interested readers can try the experiments themselves by visiting the **interactive version of the notebook**.

To the reply to all questions and comments we follow the response code: "EC" stands for "editor comment", "RC" stands for "reviewer comment", "AR" for "authors response", followed by a "diff file" between the current version and the previous revised version of our manuscript.

**1 Reply to Etienne Berthier's comments**

**EC:** *"I myself provided several (rather minor) suggestions in an annotated version of your manuscript."*

**AR:** We have done most of the modifications requested in your annotated version. Please find at the end of the reply document the "diff.pdf file" highlighting the changes. For those modifications that we chose not implement we have also a comment below.

**EC:** *"Regarding the reviewer comments about the need to clarify (or modify) the iterative calibration procedure. What about a small flowchart showing the steps of the iteration? Good help the readers."*

**AR:** We have now implemented a different method that does not involves an iteration to find a frontal ablation flux. Therefore, we did not add a flowchart and hope that the explanation in Sect. 3.3 will be sufficient to understand the method. We have also added the ideas from the comments made by Douglas Brinkerhoff (See Figure 3. and Section 3.3) and include our **Notebook** as a supplementary material.

**EC: P8L14 -** *"are they that wrong in SRTM and DEM viewfinder so that you cannot use them? I think a justification would be welcome because many reader will wonder this.*

**AR:** Water bodies in a DEM can be either flattened or set to no data points. According to the **SRTM** and **the Viewfinder Panoramas DEM3** metadata, the water bodies were flattened to a constant elevation. There may be a slight gradient on some smaller lakes, making it harder

to select an elevation value but indeed this is possible. For a large number of glaciers, however, this will, require an automatic algorithm that is able to identify a constant elevation contour and assign that elevation to a lake corresponding to a given RGI glacier outline. While this seems possible to do, it would be impeded by the dynamic changes of glacier lakes. We acknowledge that including lake-terminating glaciers is possible in principle and eventually shall be done, but would require adding substantial complexity and for this manuscript we prefer limiting our results to marine-terminating glaciers only.

**2 Reply to Douglas Brinkerhoff comments**

**RC:** *"Regarding my discussion of a missed factor in the handling of width averaged fluxes. This reply isn't entirely satisfactory. At the end of the day, the authors' stated equations are incorrect, and they should be corrected before publication: u is stated in Eq. 2 as the average cross-sectional velocity, while it is clearly not, so long as the authors allow for their modelled cross-sections to be non-rectangular. The authors should either acknowledge that their equations are an approximation that could be off by a substantial factor, or they should implement the analytical solution to u as a function of cross-section shape. The latter would be preferable, but the former is also reasonable. However, simply writing an incorrect equation and then making an appeal to antiquity ("A systematic error a very large magnitude is unlikely to have been left unnoticed") isn't defensible. The argument that by specifying the flux via integration of the upstream effective mass balance, this makes the errors in the resulting thickness go away is also not correct. Here, we have that $q = uS$, where $u$ is the depth and width average velocity given by*

$$u = \int_W \overline{u}(h(y))dy = k\overline{u}(h_0),$$  (1)

*where (in the absence of lateral drag) $k \leq 1$, with $k = 1$ for a rectangular cross section, and $k = 128/315$ for a parabolic one. This leads to the equation*

$$q = p(kf)wh_0^{n+2},$$  (2)

*where*

$$p = (2A/n + 2)(\rho g \alpha)^n$$  (3)

*$f$ is the ratio of the cross-sectional area to a rectangular one (i.e. 1 for rectangular, 2/3 for parabolic), and $w$ is the width. Solving for $h_0$ gives us*

$$h_0 = (kf)^{-1/5}(q/pw)^{1/5}$$  (4)

*For the rectangular case, of course the geometric prefactor is unity, but for the parabolic case, its around 1.3. Since this factor is so straightforward to calculate, why not just include it? It requires no separate solver, just the imposition of the correct geometric pre-factor when estimating $h_0$. If the bed shape isnt known (which it usually isnt), then the fact that the pre-factor is unknown should be included in error estimates."*

**AR:** We would like to apologize: we agreed to your comments in our previous reply but did not change the manuscript accordingly. We have now changed the text to make it clear that our approach is an approximation, and are sorry for the confusion that this oversight from our behalf may have caused.

That being said, we are thankful for your comment and agree that we should implement the proposed change in the model in the near future. Including this factor, however, would involve more changes than adding a multiplicative factor: in the case were Glens N is a known positive integer the factor can be computed (we come to the same result), it does not seem trivial for any real positive N, and currently N is a free parameter in the model (although we never used any other value than 3). Furthermore, we also support trapezoidal bed shapes in the forward dynamical model, and for the model design it was very convenient to use a general solution for u regardless of all bed shapes and just take the section into account for the flux. This, plus the fact that backwards incompatible changes in the model are not something we can take lightly, have prevented us to apply these changes for this presented manuscript.

**RC:** *"Regarding the definition of depth.*
*I think it would be better to develop the depth definition such that it could be applied to situations in which the water elevation isnt zero, e.g. for lakes. Thus, we would have that*

$$d = H_f - E_t + z_w \tag{5}$$

*where $z_w$ is the water surface. After such a definition, if there is in fact no way to measure the surface elevation of water bodies that are not the ocean, then one can explicitly set $z_w$ to zero."*

**AR:** We agree that in principle, it is possible to also apply the parameterisation to lakes. However, for technical reasons, this is not trivial, and we prefer to limit the application to marine glaciers here. See answers to editor, section 1 **EC:P8L14**

**RC:** *"Iterative calibration procedure. I think that the five steps outlined in this section are a bit confusing because some of them are done several times. Would it be possible to put this in pseudocode, as is typically done in math/CS papers? The LaTeX algorithm environment is designed for just such a task. Also, after a fair bit of fiddling with equations, I managed to convince myself that this problem does have a unique solution, and that this solution induces a unique value of $\mu^*$. Indeed, the explanation as it appears in section 3.5 is, I think, correct. However, its a little bit difficult to understand, and should also be placed in Sec. 3.3.*
*I would explain it as such: there are two equations in this model that can be used to estimate the ice flux given the thickness: Glens flow + sliding law (quintic in thickness), as can the calving law (quadratic in thickness) under the assumption that the amount of ice calved is equal to the amount delivered to the terminus. These fluxes must be equal, which leads to an equation which is quartic in thickness, the unique real-valued solution of which yields the thickness value consistent with both Glens and calving laws. Inverting either of these laws to get an equation for terminus flux and applying the computed thickness value leads to the one and only terminal flux that is consistent with both thickness estimation methods. The authors impose this specific flux in OGGM by adjusting the degree day factor $\mu^*$.*

**AR:** Thank for your ideas on how to explain the iteration procedure, we have taken some of your words and included them in our manuscript together with a new explanation of the numerical solution for the ice thickness. Please, see Section 3.3.

**RC:** *"Interestingly, being quartic, I think that the optimal thickness has an analytical solution:*

*instead of computing it directly, the authors are using a fixed-point iteration to find the roots (which I dont object to at all). In any case, a clearer description of what exactly this calibration procedure is doing mathematically would help the skeptics amongst us.*

**AR:** We have now implemented a new numerical solution for the frontal ablation flux by using two optimization tools from the **scipy** library in Python that find the roots of a function (Eq. 10 in the manuscript) in a given interval. We decided against an analytical solution for two main reasons:

- finding a numerical solution implied the least change (and copy-paste) in the existing code base, therefore minimizing the risk of creating bugs - also in the future, when we implement new formulation like you suggest above

- numerical solvers are very flexible and can virtually be applied to any other formulation of $q_{calving}$ and $q_{OGGM}$, i.e. the method will still be applicable if we use the lateral drag parameterization in OGGM or use another formulation for the calving law.

Refer to Section 3.3 of the manuscript or the **Python Notebook** for more details this method.

**RC:** *"Also worth noting: instead of adjusting the degree day factor, one could adjust other parameters (i.e. sliding law, calving parameter) such that this relationship is satisfied as well. A little bit better justification of why $\mu^*$ is the one to fiddle with is in order."*

**AR:** We agree. One could adjust, e.g., Glen's A and the sliding parameter. However, this would preclude us from using these parameters to find model solutions that agree with independent calving estimates. More importantly, the MB model calibration is based on an assumed equilibrium state (similar to the ice thickness estimation method in Farinotti et al. (2009)), such that conceptually, there is a direct link between the calving flux estimate and $\mu^*$ (as there is one between the calving flux and the ice thickness distribution). Also note that we are showing the impacts of these parameters in the frontal ablation flux in the sensitivity experiments presented in section 4.4.

**RC:** *"This can be due to two equally likely factors: solid precipitation is underestimated, or the frontal height and therefore frontal ablation is overestimated". Alternatively, k is overestimated or the calving law is wrong."*

**AR:** We agree and added this to our manuscript. [**P9L30**]

**RC:** *"In the Columbia Glacier case study, would it be possible to report the terminal velocities that OGGM predicts, as a way to see whether these are remotely consistent with observations? Columbia is, of course, very fast, and if the velocities are too low, this could lead to an overestimation of near-terminus thickness."*

**AR:** See the following answer.

**RC:** *"In Section 4.3, the authors compare the velocities produced by the model to the case where frontal ablation is not considered, which is useful. More useful, I think, would be comparing these velocities to measured ones (e.g. from GoLive) in order to get a sense of whether the physics being used by OGGM are consistent with reality."*

**AR:** Note that the velocities presented in this section are not surface velocities but average velocities computed from ice flux divided by the glacier cross-section. Annual surface velocities would have to be estimated from these values in order to compare with real data within the

[Figure]

Figure 1: **a:** Hubbard Glacier average velocities computed from ice flux divided by the glacier cross-section. **b:** Velocity structure of the Icefield Ranges, St Elias Mountains, derived from speckle tracking of ultra-fine wide RADARSAT-2 imagery from February to April 2012. Taken from Waechter, A., et al. (2015).

same time resolution. The intention of the section is to highlight the need of a frontal ablation parameterisation at the initialisation stages of the model, needed in order to create a more realistic tidewater geometry (and thus, behaviour).

During this stage, we are modeling calving glaciers under an equilibrium condition, model results presented here might not reflect the transient states that appear in observations. The usefulness of any comparison to other data (observed or modeled) is therefore limited and we wouldn't want to mislead the readers thinking both velocities (OGGM and observations) are directly comparable.

Additionally, some of the velocity maps in Alaska previously published (e.g. Burgess et al., 2013) are computed with many glaciers undergoing significant interannual velocity variability

over the observation interval and only one velocity snapshot is included in the maps, velocities might thus not represent annual average velocities.

However, for the Hubbard Glacier we carried out a "qualitative" visual assessment, comparing our velocities with estimates from Waechter, A., et al., (2015). We consider that velocities for the Hubbard Glacier (RGI60-01.14443) in Figure 1a are realistic and within the same order of magnitude than the ones reported by Waechter, A., et al., (2015), displayed in Figure 1b.

The limitation of equilibrium conditions (considered in our model, as explained above) vs. the transient state of the observations is relevant here, too. The main goal of our study is a better understanding of the parameter sensitivity of the model - and we use the results of McNabb et al., 2015 for defining what range of results we should consider realistic.

In transient runs (a natural next step), such comparisons with observed velocities can (and will) be done in a more meaningful way.

**RC:** *"In Section 4.4, the authors perform a sensitivity study over various physical parameters. This is an important step, but I think that the range of possible basal tractions is far too small. Velocities often vary between tidewater termini by two orders of magnitude. I would like to see the basal traction parameter vary over similar ranges, primarily because this parameter aliases differences in effective pressure: alternatively the sliding law used in this paper could include effective pressure explicitly, computed using height above overburden as an upper bound."*

**AR:** We have extended the range for the sliding parameter (See Fig 7). However, it should be noted that this parameter does not reflect the magnitude of the actual sliding velocity. The sliding velocity implemented in OGGM is indeed a very simple parametrisation based on (Oerlemans, 1997; Budd et al., 1979) and we think that the default value of the sliding parameter is indeed outdated. We would like to implement and test different sliding velocities parameterisations in the future. Therefore, solving for the ice thickness-calving numerically, instead of analytically, will give us plenty of room to modify all these elements inside the ice flux equation ($u$ and $u_s$) without changing significant other parts of the model code.

**RC:** *"The Discussion isn't really a discussion, it's another methods/results section in which the authors include additional observations in their inversion. I would suggest relabeling it as such."*

**AR:** While it may be unconventional to place some results in the discussion section, our intention is to show and discuss the limitations of model performance for individual glaciers can be overcome if we account for additional data. Discussing such limitations is what should be done in the discussion section, and our approach might only seem unusual because we include additional analysis. Therefore, we have decided to keep the current format.

**3    Reply to Anonymous Referee # 1**

**RC:** *"General comments.*

*First, I want to express my appreciation for the serious efforts to address the concerns is raised on the convergence. The authors now clearly show that the iterative procedure does converge to the same calving front thickness starting from various initial thickness values. The authors make clear that this iterative procedure is actually nothing else than solving a polynomial of degree 4 in two variants: with and without sliding. This also means that there is an analytical solution to the calving front ice thickness $H_f$ and thus the calving flux $F_{calving}$. This resolves my initial*

concern on the 'missing target quantity of the iterative optimisation. It is inherent in their formulation. My sincere apologies that it took me a moment to get my head around this. However, in the presented iterative scheme the $\mu^*$ calibration is entangled with the iterative determination of the calving flux $F_{calving}$. This makes the iterative procedure unnecessarily complex and blurs which variables actually determine $F_{calving}$. In my opinion, it is possible to neatly separate the calving flux problem from the $\mu^*$-calibration. If I am wrong here, please justify well and ignore my comments below. Otherwise, please disentangle these two steps to avoid needless complexity. The presentation of the procedure will strongly profit in terms of clarity. In essence, your iterative procedure boils down to setting the flux values in Eqs. (1) and (8) to be equal at the calving front. In this way, the calving flux is eliminated and it no longer needs to be determined by integrating the apparent mass balance along the glacier centreline. The solution for $H_f$ is thus independent of $\mu^*$. The resultant polynomial of degree 4 has an analytical solution and you could avoid any iterative procedure. Moreover, an analytical solution would add to the computational efficiency of OGGM. Though I prefer an analytical formulation, I refrain from urging the authors to implement the equations for the root determination. However, I urge the authors to make clear that these iterations are simply a numeric strategy to solve for the real roots of a 4th order polynomial. Clarify that this polynomial is independent of the apparent mass balance integration. You should also consider to simplify and streamline the iterative procedure."

"Here, my suggestion:
**Step 1**: Determine initial thickness $H_f^0$ guess (1/3 $E_f$ or whatever) at calving front and compute a first guess for the calving flux $F_{calving}^0$ using Eq. 8. Set i=0.

**Step 2**: Use $F_{calving}^i$ to update the ice thickness $H_f^{i+1}$ using Eq. (1) either with or without sliding.

**Step 3**: Use Eq. 8 to update the calving flux $F_{calving}^{i+1}$

**Step 4**: Set $i = i + 1$. Iterate Step 2 and Step 3 until convergence is reached.

I think the above scheme becomes much easier to implement, if you swap Eqs. (8) and (1). Then, you would only need to solve for the roots of a polynomial of degree 2 (Eq.8) to update the thickness guess in **Step 2**. The polynomial of degree 5 (Eq. 1) would then only be evaluated in **Step 1, Step 3** by inserting a the iteratively updated thickness value. Once the calving flux $F_{calving}$ is determined, you simply add it to the subsequent $\mu^*$-calibration. You could even keep the condition that $\mu^*$ should not become negative. The structure of the manuscript would need to be slightly adjusted to clarify the one-way dependence of $\mu^*$ on $F_{calving}$ and NOT vice versa."

**AR:** We have followed your advice and eliminated the iteration procedure and we now solve directly for the ice thickness. Please have a look at the beginning of this reply and Section 3.3 of the manuscript for details. We also updated our **Python Notebook** (to be included as supplementary material) for more information.

**RC:** "Specific comments.

Explain the improvement of using Eq. (8) to prescribe the calving flux for Eq. (1). Why not simply use a fixed value per region as in Huss and Farinotti (2012). What is the physically added value of the parameterisation from Oerlemanns (2005). A motivation of this choice is missing (P7L30ff). Why is this a good or even better choice?"

**AR:** Huss and Farinotti (2012) account for the calving flux of large water-terminating glaciers by reducing the ELA that yields a balanced surface mass budget by a value $\Delta_{calving}$ which is

separately defined for each RGI region but is not glacier-specific. Our approach ensures that each glacier has a frontal ablation flux compatible with its mass distribution and geometry. The MB correction of Huss and Farinotti (2012) prescribes calving, resulting in a ELA decrease of their linear mass-balance model, we compute calving and update mass-balance model parameters (here, melt). The number of glaciers of which they find the possibility for a considerable calving flux is relatively limited. Whereas our approach provides a frontal ablation flux per RGI entity. Huss and Hock (2015) have a more similar approach to our method using the same calving law from Oerlemanns and Nick (2005), the difference between our approach and theirs is that we now also invert for the frontal ice thickness. Huss and Hock (2015) estimate the thickness at the calving front by using a scaling formula relating the terminus thickness to the length of the glacier. Length variations of glaciers are, however, much more difficult to understand, as large glacier length fluctuations may arise from intrinsic climate variability. Our method relies on the thickness output of OGGM to calculate the thickness at the calving front. Finally, our motivation for using Oerlemanns and Nick (2005) parameterisation is mentioned at the introduction of our manuscript; we would like to find and test a low-cost and robust parameterisation of frontal ablation that is independent of measurements at the calving front and that is computationally efficient, since OGGM seeks to simulate past and future glacier changes on the global scale.

**RC:** *"Rethink the necessity of section 3.5."*

**AR:** We eliminated section 3.5 but retained some of the experiments that also fit the new solution method (See section 3.3.2).

**RC:** *"You could check internal consistency: Remember $F_{calving}$ used for the $\mu^*$-calibration. Run the thickness reconstruction and insert $H_f$ into Eq. (8). Do you get the same value?"*

**AR:** Yes this has been checked and added to the **Python Notebook**. We also use this consistency check in our **test suite**.

**RC:** *"Technical corrections."*

*"P7L13ff Comparing Eq. (6) and (7), it seems that q is equal to $F_{calving}$ at the calving front. You could use one variable name for the flux, either F or q."*

**AR:** Corrected: $F_{calving}$ is now named $q_{calving}$.

**RC:** *P11L2 What is the default value for k. If you used $k = 2.4\ yr^{-1}$ here, this would mean that after the regional calibration to $k = 0.66\ yr^{-1}$, the Columbia calving flux would decrease by a factor 4. This is important. In this case, the McNabb values would no longer compare well and you should reconsider your regional calibration. Please specify the default value for k and discuss.*

**AR:** There is no change in the frontal ablation flux for the Columbia glacier between the default $k$ value and $k = 0.67\ yr^{-1}$, since in this case the frontal ablation flux comes from clipping $\mu*$ and using Eq. 1. In this particular case $q_{calving}$ it is independent of $k$, this allow us to compare with the Columbia bed map computed by McNabb et, al. (2012). See section 4.4 for more information on how much the $q_{calving}$ is affected by different values of $k$.

**RC:** *P11L4 Fig.7 ! Fig.6 (please re-check all figure references.)*

**AR:** This error has been corrected.

*P13L24 '..., we increase ...' ! 'estimate is increased from ... to ...'*

**AR:** This error has been corrected.

[revised manuscript text omitted]

---

## Author Response (AR3)

**Point by point response to editor and reviewers - third revision**

We would like to thank you all for your constructive feedback on our manuscript and for your patience during this review process. We have addressed all points raised by the reviewers and hereby submit a new version of the manuscript together with a point by point reply to each of the reviewers' comments.

To the reply to all questions and comments we follow the response code: "RC" stands for "reviewer comment" and "AR" for "authors response", followed by a "diff file" between the current version and the previous revised version of our manuscript.

**1   Reply to Douglas Brinkerhoff's comments**

**RC:** *"Returning once again to the discussion of a missed factor in the handling of width-averaged fluxes, the paper needs to be specific in how the omission could affect the results, e.g. neglecting variations of u across the glacier width may lead to an overestimation of fluxes by a factor of xx, however we proceed with this approximation because it allows for easier generalization to alternative bed shapes. The authors have already written something similar in their response, but it does not enter the manuscript."*

**AR:** We agree. We added: *"We then assume that the centerline velocity is equal to the average section velocity ($\bar{u} \approx u$), which in absence of lateral drag is correct for a rectangular bed shape but isn't in the parabolic case, where we neglect the variations of the shear stress (and u) along the parabola. In the parabolic case and with N=3, this results in a section velocity overestimation of a factor 315 / 128 (approx 2.46) in comparison to the section velocity obtained by integrating the shallow-ice velocity over the parabola. We proceed with this approximation because (i) this factor cannot be computed analytically for any other non-integer value of Glen N or for other bed shapes (e.g. trapezoidal) and (ii) the uncertainties about the true shape of the bed would make the model very sensitive to this choice."*.

**RC:** *"I think that Eq. 7 still has a units error. $\int_\Omega \rho \frac{\delta h}{\delta t} dA$ has units of mass per time, yet this is equated to $q_{calving}$ which has stated units of length per time. Which is it, and is there an equivalent mistake in the code?"*

**AR:** In the code, $q_{calving}$ is converted to units of specific MB, i.e. kg m$^{-2}$ yr$^{-1}$, **here**. We have corrected this equation in the manuscript to clarify this step (see. Eq. 7 and p7L26-27).

**RC:** *"Regarding the definition of depth, I didn't mean to say that you have to include lake-terminating glaciers in the analysis, simply that it is better to include the complete derivation of a method, then simplify it later, which is to say that you can write*

$$d = H_f - E_t + z_w \tag{1}$$

*and then say that you are setting zw to zero for the rest of the paper. Would it not be valid to use this method if I started measuring water surface from some other arbitrary datum than mean sea level? I think that it would, and by using the above definition, you retain that capacity, even if you dont use it."*

**AR:** We agreed. Added to the manuscript and to Eq.9 (see p8L14-19).

**RC:** *"I wrote in my previous review: In the Columbia Glacier case study, would it be possible to report the terminal velocities that OGGM predicts, as a way to see whether these are remotely consistent with observations? Columbia is, of course, very fast, and if the velocities are too low, this could lead to an overestimation of near-terminus thickness. I didnt write this comment out of pure curiosity, but rather because I think it needs to be demonstrated that this model produces sane results for these tidewater glaciers. Thus, I think that the authors response to this point should be included in the paper somewhere."*

**AR:** Part of the previous reply has been added to sect. 4.3 (see p12L9-17).

**RC:** *"In Section 4.2, paragraph 2, the authors state that smaller glaciers change relatively more than large glaciers for a specified frontal ablation. I think that the different scale of the glaciers overwhelm potential differences in dynamics, making these results less interesting than they could be. To me, its rather obvious that specifying a given and fixed calving flux would have a larger relative effect on a smaller glacier, because its a relatively larger portion of the mass budget. A more interesting experiment might be to specify a range of frontal ablation fluxes as fractions of the total accumulation, e.g. $q_{calving} = f \int_{\Omega} p_f P_{solid} dA$ for $f \in [0;0:5]$. Thus the frontal ablation would naturally scale with the glacier size and we could see the result of the differences in physics between small and large glaciers."*

**AR:** We agree. Figure 5 has been modified such that the x-axis shows frontal ablation fluxes as a fraction of the total accumulation over each glacier. Doing this analysis we realised that the effect of accounting for frontal ablation in the glacier volume does actually not depend on the glacier size, and the relationship is more complicated than what we originally thought. However, this effect is systematic in that accounting for frontal ablation in the MB always results in an increase on the glacier volume. Figure 5a shows that even if the frontal ablation fraction is only 0.14 of the total accumulation over a glacier, this glacier volume can be underestimated by up to 20% when ignoring this extra loss of ice in the MB. We have changed the text in the manuscript to highlight this new (and now correct) finding. Please see sect. 4.2.

**2 Reply to Anonymous Referee # 1**

**RC:** *"P6L14 Check the units of the sliding parameter. I think that they should be m2s1Pa3."*

**AR:** Corrected. Please see p6L14.

**RC:** *"P8L7 Please remove the k-value as given by Oerlemanns and Nick (2005). It gives rise for confusion later on where you mention default values (P10L21)."*

**AR:** Removed, we have specify now that the 2.4 is the default setting for this parameter in OGGM. Please see p8L11.

**RC:** *"P9L18-19 Please remove the first argument against implementing an analytical solution because the history of the review process will not be visible to a wide audience."*

**AR:** Removed. Please see p9L20-23.

**RC:** *"P10L5-10 You should remember the reader about the calibration of the calving factor k in*

*Sect. 2.3. This calibration defines the default values for k, A and $f_s$. Give their values or refer to Table 1."*

**AR:** A reference to Table 1 and the calibration experiments has been added. Please see p10L11-12.

**RC:** *"P10L28-32 I think that it is if high interest to further expand on the Columbia Glacier case study. Please also compare your results to the consensus estimate of glacier ice thickness from Farinotti et al. (2019). Is there a qualitative difference? This is not a lot of work. Just add an extra line in Fig. 4c and add 2-3 sentences of explanation. To put this comment in a question: Why should we prefer the OGGM thickness values over the consensus thickness estimate for marine terminating glaciers in Alaska?".*

**AR:** This has been added. Please see Fig.4c. and section 4.1 (p11L2-4).

**RC:** *"P11L30-P12L3 This passage is confusing because it evokes alternative choices in the methodology, which were not pursued. Please consider to transfer it to the methodology or remove it.".*

**AR:** Thanks for your suggestion. We have now changed the text to illustrate our point better. Please see p12L4-8.

**RC:** *"P12L29 I understand that you define the equally good parameter sets on the basis of Fig. 7. From Table 1, I see the following ranges: A from 2.4 via 2.41 to 2.7 $\times 10^{-24}$ $s^{-1}$ $Pa^{-3}$, k varies between 0.63 and 0.67, and $f_s$ is only turned off and on. The creep parameter A is therefore only increased. Are the two values 2.4 and 2.41 different settings? "*

**AR:** Yes both settings are different, configurations 5 and 8 are derived from estimating the intercept between the red and blue lines in Fig.7b, with the Alaska frontal ablation estimate from McNabb, et al. (2015). Configurations 5 and 8 have an intercept very close to the default value of $GlenA$ in OGGM. Config. 5; A =2.4057 $\times 10^{-24}$ $s^{-1}$ Pa and Config. 8; A = 2.4018 $\times 10^{-24}$ $s^{-1}$. Both configurations have the corresponding $k$ values derived with A = 2.4 $\times 10^{-24}$ $s^{-1}$ $Pa^{-3}$ (default set up in OGGM) from the previous experiment, therefore it is expected that both intercepts are so close to the default Glen A value, but they are written in table 1 as a different configuration. For making the table shorter we only take into account two decimal points but in the code the complete values are taken (see the **documentation** for the exact values on each configuration).

**RC:** *"Why did you not decrease this value. The MacNabb frontal ablation uncertainty would allow a total range of roughly 1.5 to 4.0$\times 10^{-24}$ $s^{-1}$ $Pa^{-3}$. The sliding parameter $f_s$ is only decrease to zero, which hardly affects the frontal ablation (as visible from Fig. 7c). Again, the associated uncertainty in frontal ablation would allow a larger range up to 20.0 $\times 10^{-18}$ $s^{-1}$ $Pa^{-3}$. Finally, the k-range seems very narrow. You only use the two calibrated values for the cases with and without sliding. Again the frontal ablation range from MacNabb et al. (2015) would allow a range from 0.5 to 0.75. In summary, the question is why you chose such a narrow and selective parameter range? Please motivate your choice or adjust."*

**AR:** We have added 5 more configurations to Table 1. These configurations were obtained from finding the intercepts between OGGM frontal ablation estimates and the lower and upper error provided by MacNabb et al. (2015). For the $k$ and Glen $A$ parameters, we only account

for the lowest and highest parameter value that intercepts the uncertainty range of MacNabb et al. (2015) (see cross marks in Figure 7. a and b). For Figure 7c and the sliding parameter, we only account for the intercept to the lower uncertainty limit since the upper bound will be a sliding parameter $f_s$ equal to zero and that configuration is already configuration number 2.

**RC:** *"P12L29-P13L6 I do miss a description of the sensitivity results due to changes in all parameters. You only say that the creep and sliding parameters are most influential. What is the relative effect on regional ice volume. From Fig. 8, I loosely infer ± 5%. How does this sensitivity compare to the uncertainty of other reconstruction approaches. The 5% value seems optimistic and it is certainly linked to your choices of the parameter ranges."*

**AR:** We agree. We now increase the range of possible configurations by including the uncertainty range of the value provided by Mc Nabb et al (2015), and compare our estimates to the consensus estimate from Farinotti. et al., (2019). Our own uncertainty estimates are therefore less optimistic, and we find that the consensus estimate is 27.58% lower than our mean regional volume estimate after accounting for calving (see sect 4.5 and Discussion p13L30-31).

**RC:** *"Most parts of the discussion section focuses on the importance of correct geometric information near the glacier terminus. This is certainly valuable but I think you should use this section to also compare your approach with other reconstruction estimates (e.g. Farinotti et al., 2019). In my view, your results will show a clear improvement. Moreover, you should also discuss the choices you made in the sensitivity analysis."*

**AR:** We agree and have now added an extra panel on Fig. 9 (b) where we compare both configurations (default set up in OGGM) and when correcting for the terminus geometry with volume estimates from Farinotti et al., 2019, derived from the consensus on the thickness distribution for these 36 glaciers. Please see Discussion p15L22-27 and Fig 9b.

**RC:** *"P14L6 What errors do you refer to here? I think you refer to the width and depth correction that is described thereafter."*

**AR:** Added. Please see p14L19-20.

**RC:** *"P15L14 Remove the parenthesis and reformulate. This information is decisive for the sentence. "*

**AR:** Corrected. Please see p16L8.

**RC:** *"Fig. 4c Could you add the result for Columbia Glacier from Farinotti et al., (2019) in panel (c). It is of great interest to the community how their consensus estimate does perform on tidewater glaciers. "*

**AR:** Added. Please see Fig.4c.

**RC:** *" Fig. 5b The sensitivity of the inferred $\mu^*$ on the calving factor k indicates that $\mu^*$ is set to zero for most of the smaller glacier (100 $km^2$) even for the default k value. This issue needs discussion in Sect. 5.*

**AR:** Figure 5b describes the changes of the glacier temperature sensitivity ($\mu^*$) per frontal ablation value assigned to the mass balance from a given range (0 - 5 $km^3yr^{-1}$). We are not estimating a frontal ablation flux in this section as stated on p11.L10 and we do not calculate

a frontal ablation flux with the model. The $k$ parameter plays no role on this figure. Note that the x-axis of the figure was changed following the suggestion of the other reviewer, and the misunderstanding should now be prevented.

**References:**

[revised manuscript text omitted]

---

## Author Response (AR4)

**Response to editor last revision**

Thank you, we did as suggested. The changes are highlighted in the "diff file" below.

[revised manuscript text omitted]